# Trans-EnV: A Framework for Evaluating the Linguistic Robustness of LLMs Against English Varieties

**Jiyoung Lee**[1]*, **Seungho Kim**[2]*†, **Jieun Han**[1], **Jun-Min Lee**[1]
**Kitaek Kim**[3], **Alice Oh**[1], **Edward Choi**[1]
[1]KAIST     [2]Suresoft Technologies     [3]Seoul National University
[1]{jiyounglee0523, jieun_han, ljm565, edwardchoi}@kaist.ac.kr
[1]alice.oh@kaist.edu     [2]shkim3@suresofttech.com     [3]kitaek@snu.ac.kr

## Abstract

Large Language Models (LLMs) are predominantly evaluated on Standard American English (SAE), often overlooking the diversity of global English varieties. This narrow focus may raise fairness concerns as degraded performance on non-standard varieties can lead to unequal benefits for users worldwide. Therefore, it is critical to extensively evaluate the linguistic robustness of LLMs on multiple non-standard English varieties. We introduce Trans-EnV, a framework that automatically transforms SAE datasets into multiple English varieties to evaluate the linguistic robustness. Our framework combines (1) linguistics expert knowledge to curate variety-specific features and transformation guidelines from linguistic literature and corpora, and (2) LLM-based transformations to ensure both linguistic validity and scalability. Using Trans-EnV, we transform six benchmark datasets into 38 English varieties and evaluate seven state-of-the-art LLMs. Our results reveal significant performance disparities, with accuracy decreasing by up to 46.3% on non-standard varieties. These findings highlight the importance of comprehensive linguistic robustness evaluation across diverse English varieties. Each construction of Trans-EnV was validated through rigorous statistical testing and consultation with a researcher in the field of second language acquisition, ensuring its linguistic validity. Our code and datasets are publicly available. [4]

## 1 Introduction

Large Language Models (LLMs) [1, 64, 27] have shown impressive performance, even surpassing humans on several tasks [44, 34, 24]. However, most evaluation benchmarks are written in Standard American English (SAE), overlooking the rich diversity of English varieties. English is spoken in a wide range of *varieties*, including regional dialects and forms used by non-native speakers [10]. This narrow linguistic focus may raise fairness concerns, as LLMs tend to underperform on non-standard varieties [78, 7], potentially leading to unequal benefits for global users. Therefore, assessing LLM performance in multiple English varieties is essential to ensure its fairness.

While several datasets have been introduced to evaluate the robustness of LLMs to varieties [8, 17, 58, 36], they remain limited in size, variety coverage, and task diversity, making them inadequate for

---

*Equal Contribution.

†Work done while at KAIST.

[4]Code: https://github.com/jiyounglee-0523/TransEnV

  Dataset: https://huggingface.co/collections/jiyounglee0523/transenv-681eadb3c0c8c f363b363fb1

39th Conference on Neural Information Processing Systems (NeurIPS 2025) Track on Datasets and Benchmarks.

comprehensive evaluation. To rigorously assess the linguistic robustness of LLMs, it is necessary to evaluate on existing benchmark datasets across a diverse range of English varieties. This, in turn, requires a framework capable of automatically converting SAE benchmarks into any desired target variety. Although prior studies have proposed such transformations [41, 56, 28, 79, 80], these approaches often suffer from scalability challenges, lack of expert knowledge, or failure to adequately capture linguistic diversity.

To this end, we introduce **Trans-EnV**, a framework that automatically transforms SAE datasets into a desired target variety. We focus on two widely studied types of variety: regional dialect and English as a Second Language (ESL) English [33, 6, 60]. Regional dialects, henceforth called as *dialects*, refer to geographically localized varieties of English, such as Scottish or Irish English. ESL English refers to language produced by speakers whose first language is not English. We begin by collecting linguistic features from expert-curated resources and large-scale ESL corpora to ensure rigorous and accurate information. Then, for each feature, we create transformation guidelines that specify the steps to apply the feature to an SAE sentence. Then we use an LLM to transform SAE sentences by following the guidelines. An overview of Trans-EnV is provided in Figure 1. LLMs are used both for the guideline generation and the sentence transformation, making Trans-EnV both labor-efficient and scalable across datasets. To ensure linguistic validity, we consulted a researcher in the field of second language acquisition throughout the entire development, aligning our methodology with established linguistic theory and practice.

We translate six widely used benchmark datasets into 38 varieties consisting of 18 dialects and 20 ESL English varieties. We evaluate seven state-of-the-art LLMs on the transformed benchmarks and observe that model performance generally degrades across most varieties, with particularly pronounced drops in ESL English (12.5% and 46.3% performance drop at maximum for each dialect and ESL English). In addition, we find that linguistic robustness is notably weaker on tasks that require reasoning. Models specialized in reasoning tend to be more robust than others, suggesting that strong reasoning capabilities may contribute to improved robustness.

Our contributions are summarized as follows:

- We introduce Trans-EnV, a framework that automatically transforms SAE-written datasets into specified target English varieties. By leveraging an LLM for transformation, our framework is both labor-efficient and scalable across multiple datasets.
- Trans-EnV is grounded in expert-curated linguistic resources, and its construction was validated through rigorous statistical testing and consultation with a researcher in the field of second language acquisition, ensuring its linguistic validity.
- We conduct extensive experiments by transforming six widely used benchmark datasets into 38 varieties and evaluating seven LLMs. The results show that LLMs exhibit notable weaknesses in handling non-standard English varieties, particularly in ESL English.

## 2   Related Work

**English Variety Disparity in LLMs.**   English variety refers to various forms of English used across different regions, communities, or learner groups, including both Standard American English and non-standard forms. The non-standard forms include regional dialects (*e.g.*, Scottish English) and ESL learner usages (*e.g.*, Arabic English) [45, 29]. Despite such diversity in English, LLMs do not perform well particularly in non-standard English in the form of dialects [35, 62, 13, 70, 39] and ESL learners' usages [46, 40]. For instance, in non-SAE settings, LLMs underperform in tasks such as language generation and understanding [13, 28, 62, 46], reasoning capability [41, 77] and instruction following [19]. In addition, LLMs respond in a more stereotyping, demeaning, unnatural, and condescending manner to under-represented varieties [19, 31, 40]. Zhou [78] showed that LLMs have shown persistent biases against non-SAE particularly in tasks involving toxicity.

**English Variety Dataset.**   Previous work on transforming datasets for varieties typically uses one of three methods: (i) manual curation [41, 56], (ii) LLM generation [28], and (iii) rule-based transformation [79, 80]. The first approach involves humans manually creating the entire dataset. While this method can produce high-quality results, recruiting qualified human annotators is resource-intensive, and scaling to multiple English varieties and datasets is challenging. The second approach relies entirely on LLMs to generate varieties. While this method is scalable and convenient, several

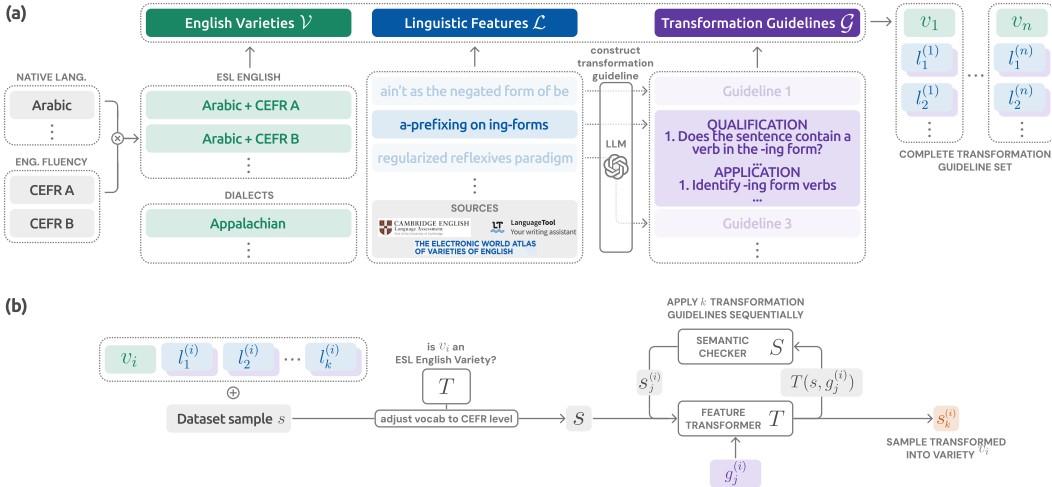

Figure 1: Overview of Trans-EnV. (a) *Data Collection* and *Transformation Guideline Generation*: We gather English varieties and their associated linguistic features from linguistic literature and large-scale corpora. For each feature, we construct a transformation guideline that defines the procedure for applying the feature. (b) *Transformation into Target Variety*: Given an SAE sentence $s$ and a target variety $v_i$, the semantic checker $S$ and feature transformer $T$ LLMs transform $s$ by sequentially applying the features of $v_i$ by following guidelines.

studies have highlighted the limitations of LLMs in accurately reproducing under-represented varieties of English[46, 68, 4, 18], underscoring the risk of relying solely on LLMs. The third approach employs deterministic, rule-based transformations to transform SAE sentences into targeted varieties. However, this method demands substantial human effort to manually craft transformation rules for each linguistic feature. Moreover, it falls short in capturing the full spectrum of linguistic variation, including lexical choices and pragmatic nuances. As a result, these transformations tend to be context-specific and challenging to generalize, limiting their applicability across different domains and language varieties [61, 28]. In contrast, our approach integrates expert-curated resources with the linguistic capabilities of LLMs to construct a robust framework that captures diverse and accurate language expressions across varieties, while ensuring both linguistic validity and scalability.

## 3   Trans-EnV: A Framework for Transforming SAE into Varieties

Constructing Trans-EnV consists of three main steps: (i) data collection, (ii) generation of transformation guidelines, and (iii) transforming SAE sentences into the target English variety. In the data collection phase, we compile a set of varieties $\mathcal{V} = \{v_1, \ldots, v_n\}$, where $n$ denotes the total number of varieties. Each variety is associated with a set of linguistic features, which we refer to as *features* for brevity. Let $\mathcal{L} = \{l_1, \ldots, l_m\}$ denote the complete set of unique features across all varieties where $m$ is the total number of features, and let $\mathcal{L}_{v_i} = \{l_1^{(i)}, \ldots, l_k^{(i)}\} \subset \mathcal{L}$ represent the subset of features specific to variety $v_i$, where $k$ is the number of associated features. During the transformation guideline generation phase, we construct a guideline $g_j$ for each feature $l_j$ specifying the operations required to apply $l_j$ to a given sentence. The set of all guidelines is denoted by $\mathcal{G} = \{g_1, \ldots, g_m\}$, with $\mathcal{G}_{v_i} = \{g_1^{(i)}, \ldots, g_k^{(i)}\} \subset \mathcal{G}$ denoting the subset of guidelines corresponding to $v_i$, following the notation above. In the final transformation stage, we convert SAE sentences into the target variety $v_i$ by sequentially applying each feature in $\mathcal{L}_{v_i}$ following the corresponding guidelines in $\mathcal{G}_{v_i}$.

Since ESL English is influenced by both the learner's proficiency and native language (L1) [43, 47, 63], our framework considers both factors. To address the vocabulary limitations common among English learners, we add an initial step that simplifies advanced words into more accessible synonyms or phrases. Figure 1 provides an overview of Trans-EnV.

### 3.1 Data Collection

**Dialect.** We utilize the Electronic World Atlas of Varieties of English[5] (eWAVE) [37], a comprehensive database that documents 235 linguistic features across 77 varieties of English. This dataset was compiled by 84 professional linguists and is grounded in 175 peer-reviewed publications. Each variety in eWAVE is annotated for every feature using a four-level scale indicating the degree of presence. Among the 77 varieties, some are English-based creoles and pidgins, which, despite sharing vocabulary and structural elements with English, have diverged significantly and are considered distinct languages [69, 3]. To systematically distinguish English dialects from these non-English varieties, we apply K-Nearest Neighbors (KNN) clustering on the 77 varieties each represented by 235 linguistic features, treating them as input embeddings. We then select clusters that contain widely recognized English dialects, such as Australian English and Scottish English. This process yields 18 dialects. The appropriateness of this subset for our research scope was verified by the specialist in second language acquisition, with the remaining varieties considered outside the intended focus. For each dialect $v_i$, we define its features $\mathcal{L}_{v_i}$ as those annotated with the highest level of presence in eWAVE. Additional details on eWAVE, the clustering procedure, and the selected dialects are provided in Appendix C.

**ESL English-Proficiency.** We adopt the Common European Framework of Reference for Languages (CEFR) [48] as our indicator of English proficiency. CEFR is a widely used standard that defines six proficiency levels (A1–C2). For our purposes, we focus on the three higher categories—A (Basic), B (Intermediate), and C (Proficient)—as finer-grained distinctions often lead to overfitting in AI applications [20, 38]. Given that level C closely approximates native English proficiency, which LLMs are generally capable of, we focus on levels A and B. We collect features from the English Grammar Profile (EGP) [54], which catalogs 1,222 features mapped to CEFR levels by systematically analyzing a large corpus containing millions of texts written by ESL learners from diverse L1 backgrounds and proficiency levels [55]. Each feature is presented in the form of a "can-do" descriptor (*e.g.*, *can use adjective phrases to modify nouns* at B2), which reflects the general grammatical abilities expected at each CEFR level. To simulate a target CEFR level, we exclude "can-do" features associated with higher levels— for example, to transform a sentence into level A, we identify and remove all B-level and C-level "can-do" features present in that sentence. Thus, features defining each CEFR level are derived from those of the higher levels as removal targets.

**ESL English-L1.** Existing linguistic studies on the influence of L1 typically examine fewer than six morphemes per language, resulting in a limited set of L1-specific features across languages. To address this gap, we empirically derive L1-specific features through controlled experiments, following established methodologies in linguistics [32, 47]. To disentangle L1-specific features from those of general second-language acquisition, features should be extracted from essays by learners within the same proficiency level, ensuring that observed features only reflect L1 influence [32]. Therefore, we need learner essays annotated with both L1 and CEFR level to conduct the controlled analysis.

We use three open-source ESL learner corpora: CLC-FCE [72], ICLE [25], and EFCamDat [21], all of which contain English essays written by learners, annotated with their native language. Among these, only EFCamDat includes CEFR proficiency annotations. We use GPT-4o mini[6] to predict CEFR levels as pseudo-proficiency indicators for CLC-FCE and ICLE. From the corpora, we select 10 L1s with sufficient data: Arabic, Chinese-Mandarin, French, German, Italian, Japanese, Portuguese, Russian, Spanish, and Turkish. We apply an automatic grammar checker[7] to each sentence to identify grammatical features, which are then grouped into higher-level categories. For each L1 and CEFR level, we compute feature frequencies and conduct statistical t-tests to identify features significantly associated with specific L1s and CEFR level ($p < 0.05$). On average, 10 distinct features were identified for each L1 at each CEFR level. We confirmed that the extracted features align with prior linguistic findings [47], and their validity was verified by native speakers of Spanish, French, Chinese-Mandarin, and Italian as well as a specialist in second language acquisition.

For ESL English, $\mathcal{L}_{v_i}$ combines feature from the target CEFR level the corresponding L1. For example, for variety defined by CEFR level A and Arabic as the L1, the feature set includes both

---

[5] https://ewave-atlas.org/

[6] Model version: gpt-4o-mini-2024-07-18. The model achieves 77.3% accuracy on CEFR-SP [2] for three-level CEFR classification.

[7] https://pypi.org/project/language-tool-python/

CEFR level A features and Arabic-specific features at CEFR level A. We verified that features from CEFR and L1s do not conflict. Full experimental details and summaries of extracted features are provided in Appendix C.2.

## 3.2 Transformation Guideline Generation

As LLMs often fail to apply features to SAE sentences when prompted with feature names alone, it is essential to provide explicit, well-defined transformation guidelines and enforce step-by-step execution. Therefore, we generate a transformation guideline $g_j$ for each feature $l_j$, which outlines a detailed, step-by-step procedure for applying $l_j$ to a given sentence. Each guideline consists of two steps: *Qualification* and *Application*. The *Qualification* step determines whether the feature is applicable to the sentence. For instance, for the feature *'She/her used for inanimate references'*, this step verifies the presence of an inanimate referent and a pronoun that refers to it in the sentence. The *Application* step provides detailed instructions to implement the transformation, *e.g.*, identifying the inanimate referent and replacing its corresponding pronouns with *she* or *her*. We use GPT-4[8] to generate these guidelines via one-shot prompting. All generated guidelines were reviewed by the researcher in the second language acquisition and were deemed appropriate for use. Further details on generation configuration, prompts used, and examples are provided in Appendix C.3.

## 3.3 Transforming into English Varieties

Given an SAE sentence $s$ and a target variety $v_i$, we transform $s$ using the associated guideline set $\mathcal{G}_{v_i}$. We utilize a feature transformer model $T$, which applies each guideline to $s$, and a semantic checker model $S$, which verifies whether the transformed sentence preserves the original meaning. Both $T$ and $S$ can be any AI model capable of interpreting and executing the provided guidelines.

For ESL English varieties, vocabulary replacement is a crucial step due to the limited lexical range of English learners. To ensure that the vocabulary aligns with the target CEFR levels, we compile a vocabulary-to-CEFR mapping from the Oxford 5000[9] and supplementary word lists,[10] resulting in 23,411 labeled words. For words that are not covered by the list, we use GPT-4o to provide their CEFR levels. Simply replacing all higher level words with target level words is not an optimal transformation strategy. Analyzing the CEFR-labeled English text dataset [11] showed that texts labeled as CEFR A and B levels contained a small proportion of higher-level words, up to 14.3% and 9.6% at the 90th percentile, respectively. Based on this observation, we allow up to 15% of higher-level vocabulary in transformed texts to reflect realistic ESL proficiency. We replace high-level words using $T$ as the first step of transformation. In cases where the vocabulary could not be sufficiently simplified (*e.g.*, complex questions from the MMLU professional law), we exclude those samples from the final dataset, as they were considered too difficult for ESL learners at the target level. Table 13 in Appendix C.4.1 presents the final dataset sizes and the ratio of successful transformations. Further details and examples of vocabulary transformation are provided in Appendix C.4.1.

Next, for both dialect and ESL English, we randomly shuffle features in $\mathcal{L}_{v_i}$ and apply them sequentially. When applying a feature $l_j^{(i)}$, $T$ determines whether the *Qualification* condition specified in $g_j^{(i)}$ is satisfied. If the condition is met, $T$ performs the transformation following the *Application* step, producing a transformed sentence. The transformed sentence is then passed to the next feature $l_{(j+1)}^{(i)}$. If the condition is not satisfied, the feature is skipped. After each transformation, $S$ verifies whether the original and transformed sentences preserve the same meaning. Only transformations that pass this semantic check are retained and used in the subsequent steps. Full prompts used in the transformation process and examples of the transformation procedure are provided in Appendix C.5.

## 3.4 Analysis of Trans-EnV

We applied Trans-EnV to six benchmark QA datasets, three knowledge-based datasets: MMLU [30], ARC [11], TruthfulQA [42], and three reasoning-based datasets: GSM8K [12], HellaSwag [74],

---

[8]Model version: `gpt-4-0613`

[9]`https://www.oxfordlearnersdictionaries.com/wordlists/oxford3000-5000`

[10]`https://www.oxfordlearnersdictionaries.com/topic/`

[11]`https://www.kaggle.com/datasets/amontgomerie/cefr-levelled-english-texts/data`

Table 1: Average number of features applied per sample and proportion of transformed samples

|  | MMLU | ARC | TruthfulQA | GSM8K | HellaSwag | WinoGrande |
|---|---|---|---|---|---|---|
| Dialect | 1.81 / 71.4% | 1.67 / 69.7% | 1.44 / 63.0% | 1.61 / 64.2% | 2.52 / 83.4% | 3.22 / 92.1% |
| ESL English | 2.41 / 92.7% | 2.48 / 94.5% | 2.03 / 88.4% | 2.63 / 95.1% | 2.64 / 94.9% | 2.76 / 97.3% |
| Total | 2.12 / 82.6% | 2.09 / 82.7% | 1.75 / 76.4% | 2.15 / 80.5% | 2.58 / 89.4% | 2.98 / 94.8% |

Table 2: Transformation examples by Trans-EnV with sequential application of two features.

| | | |
|---|---|---|
| | **SAE** | There are 66 fish in the fish tank. One-third of the fish have red stripes . . . fish have red stripes and blue stripes? |
| Example 1 | **Feat. 1** | Regularization of plural formation: extension of -s to StE irregular plurals |
| | **Transf. 1** | There are 66 fishs in the fish tank. One-third of the fishs have red stripes . . . fishs have red stripes and blue stripes? |
| | **Feat. 2** | Existential / presentational there's/there is/there was with plural subjects |
| | **Transf. 2** | There's 66 fishs in the fish tank. One-third of the fishs have red stripes . . . fishs have red stripes and blue stripes? |
| | **SAE** | Joe has twice as many cars as Robert. He sells 20% these ones and gives away twice as many cars as the number . . . |
| Example 2 | **Feat. 1** | Usage of a singular noun when a plural form is required |
| | **Transf. 1** | Joe has twice as many car as Robert. He sells 20% this one and gives away twice as many car as the number . . . |
| | **Feat. 2** | Omission of a preposition |
| | **Transf. 2** | Joe has twice _ many car Robert. He sells 20% this one and gives away twice _ many car _ the number . . . |

WinoGrande [59]. We used Gemma-2-27B-Instruct [66] as the feature transformer model $T$, and LLaMA-3.3-70B-Instruct [26] as the semantic checker model $S$. Each dataset is transformed into total of 38 varieties—18 dialects and 20 ESL English.

**Transformation Coverage and Intensity.** Table 1 reports the average number of features applied per sample and the overall proportion of transformed samples across datasets. On average, around two features were applied per sample. Given that most samples are relatively short, consisting of one or two sentences, this level of transformation is considered reasonable. In most cases, over 80% of the samples were modified and ESL English samples exhibited a higher rate of transformation than dialect. This may be attributed to ESL English features being more closely tied to everyday usage than those of dialects. We found that untransformed samples are significantly short or simple in structure, such as *"Let p = (1, 2, 5, 4)(2, 3) in $S_5$. Find the index of <p> in $S_5$."* or *"What is 'coring'?"*, which left little room for transformation. We provide examples of transformed sentences in Table 2. Detailed statistics for each variety within each dataset are provided in Appendix C.6.

**Human Evaluation.** We evaluate the effectiveness of our framework using six different models as Feature Transformer ($T$): LLaMA-3.1-8B-Instruct [15], Gemma-2-27B-Instruct [66], Gemma-3-27B-Instruct [65], Qwen2.5-32B-Instruct [67], GPT-4 [49], and GPT-4.1-mini [51].[12] For human evaluation, we focused on recruiting participants with strong proficiency in English grammar, as the evaluation guidelines involved grammatical terminology and sentence structure analysis. We recruited graduate students who have had formal coursework in English linguistics or grammar. These candidates

Table 3: Human evaluation on transformed sentences from six different Feature Transformers.

| Feature Transformer | Q1 | Q2 | Final |
|---|---|---|---|
| LLaMA-3.1-8B | 25 | 14 | 14 / 25 (56%) |
| Gemma-2-27B | 25 | 23 | 23 / 25 (92%) |
| Gemma-3-27B | 25 | 24 | 24 / 25 (95%) |
| Qwen2.5-32B | 24 | 24 | 23 / 25 (92%) |
| GPT-4 | 25 | 24 | 24 / 25 (96%) |
| GPT-4.1-mini | 25 | 24 | 24 / 25 (96%) |
| Total | 149 | 133 | 132 / 150 (88%) |

were asked to complete a grammar pre-test designed to assess their understanding of key concepts relevant to the evaluation task. Based on their performance, we selected three individuals who scored the highest.

Each model generated 25 transformed outputs, resulting in 150 samples per annotator. Outputs were evaluated on two criteria: (Q1) whether the model correctly followed the Qualification and Application steps specified in the guidelines, and (Q2) whether the transformed sentence preserved the original meaning. Table 3 presents the results. A sample is considered valid if it received

---

[12]Model versions: `gpt-4-turbo-2024-04-09`, `gpt-4.1-mini-2025-04-14`

Table 4: Benchmark results of seven models on dialect varieties. Values in blue cells are those that performed better than the original.

| | | Orig. | Mean | AAVE | AppE | AuE | AusVE | BahE | TdCE | EAngE | IrE | Manx | NZE | NfE | NE-Eng | OzE | ScE | SE-AmE | SE-Eng | SW-Eng | WeE |
|---|---|---|---|---|---|---|---|---|---|---|---|---|---|---|---|---|---|---|---|---|---|
| MMLU | Qwen2.5-72B | 84.6 | 82.6 | 83.2 | 82.5 | 83.1 | 83.9 | 82.8 | 83.7 | 82.1 | 81.0 | 82.3 | 82.3 | 81.1 | 83.5 | 82.0 | 83.2 | 81.8 | 84.2 | 82.9 | 81.3 |
| | DeepSeek-R1-70B | 84.3 | 84.8 | 85.4 | 84.6 | 85.4 | 85.6 | 84.5 | 86.5 | 84.5 | 83.7 | 85.1 | 83.1 | 83.4 | 85.5 | 84.4 | 85.5 | 84.2 | 86.1 | 85.2 | 83.8 |
| | Llama-3.3-70B | 84.3 | 82.3 | 82.6 | 82.0 | 82.7 | 82.7 | 83.7 | 82.7 | 83.7 | 80.5 | 82.4 | 83.6 | 80.5 | 82.6 | 81.3 | 82.6 | 81.4 | 83.3 | 82.2 | 81.0 |
| | gemini-2.0-flash | 86.0 | 83.7 | 84.2 | 83.7 | 87.4 | 84.6 | 83.2 | 84.9 | 82.9 | 81.8 | 83.8 | 83.0 | 81.3 | 84.4 | 82.9 | 84.3 | 82.9 | 85.0 | 84.1 | 82.6 |
| | gemini-2.5-pro | 90.9 | 88.7 | 89.2 | 88.6 | 89.6 | 89.9 | 88.2 | 90.1 | 88.3 | 87.0 | 88.7 | 88.4 | 87.2 | 89.4 | 88.0 | 89.4 | 88.0 | 90.3 | 89.3 | 87.9 |
| | gpt-4o-mini | 76.3 | 74.6 | 75.1 | 74.1 | 75.6 | 75.7 | 74.1 | 75.3 | 74.1 | 73.3 | 74.6 | 74.8 | 72.4 | 75.4 | 73.9 | 75.4 | 74.1 | 75.7 | 74.8 | 73.7 |
| | o4-mini | 89.3 | 87.3 | 87.7 | 87.0 | 88.0 | 88.3 | 87.2 | 88.7 | 86.9 | 85.9 | 87.6 | 87.1 | 85.7 | 87.8 | 86.5 | 88.1 | 86.9 | 88.9 | 87.4 | 86.3 |
| ARC | Qwen2.5-72B | 96.0 | 95.0 | 95.0 | 95.6 | 95.6 | 95.9 | 95.1 | 95.6 | 94.7 | 94.8 | 95.0 | 95.1 | 93.8 | 95.3 | 94.5 | 95.2 | 95.1 | 95.6 | 95.0 | 94.2 |
| | DeepSeek-R1-70B | 95.3 | 94.8 | 94.3 | 94.7 | 95.1 | 95.1 | 95.5 | 94.6 | 94.6 | 94.3 | 94.6 | 95.5 | 93.6 | 95.1 | 94.4 | 95.6 | 94.4 | 95.3 | 95.0 | 94.3 |
| | Llama-3.3-70B | 95.2 | 94.1 | 94.6 | 94.5 | 94.8 | 94.9 | 94.1 | 95.1 | 93.9 | 93.3 | 94.4 | 94.0 | 93.3 | 94.6 | 93.3 | 94.5 | 93.9 | 95.1 | 94.0 | 92.6 |
| | gemini-2.0-flash | 95.9 | 94.9 | 96.0 | 87.7 | 95.9 | 96.0 | 95.2 | 95.2 | 95.3 | 95.3 | 94.9 | 95.7 | 93.9 | 95.6 | 94.8 | 95.4 | 94.7 | 95.8 | 95.9 | 94.5 |
| | gemini-2.5-pro | 97.5 | 96.8 | 96.8 | 96.9 | 96.8 | 97.1 | 97.3 | 97.0 | 96.5 | 96.9 | 96.8 | 97.5 | 95.3 | 96.7 | 96.5 | 97.1 | 96.2 | 97.4 | 96.6 | 96.4 |
| | gpt-4o-mini | 92.2 | 91.2 | 91.5 | 92.0 | 91.0 | 92.4 | 91.4 | 91.5 | 90.7 | 90.5 | 91.6 | 91.1 | 89.7 | 91.4 | 90.7 | 92.4 | 91.0 | 91.8 | 91.3 | 90.0 |
| | o4-mini | 97.0 | 96.3 | 96.6 | 95.2 | 96.6 | 96.8 | 96.1 | 96.8 | 96.5 | 96.8 | 96.6 | 97.2 | 94.8 | 96.0 | 96.2 | 96.8 | 95.7 | 96.8 | 96.6 | 95.1 |
| TruthfulQA | Qwen2.5-72B | 77.0 | 77.1 | 77.0 | 77.7 | 78.5 | 76.0 | 77.1 | 78.5 | 76.5 | 75.8 | 76.7 | 76.7 | 76.3 | 77.8 | 76.7 | 77.5 | 77.4 | 77.2 | 78.8 | 75.2 |
| | DeepSeek-R1-70B | 72.0 | 70.9 | 70.9 | 71.6 | 71.6 | 71.5 | 71.0 | 72.8 | 70.5 | 68.4 | 70.0 | 71.6 | 68.3 | 70.9 | 69.5 | 72.6 | 70.9 | 72.0 | 71.1 | 71.0 |
| | Llama-3.3-70B | 71.2 | 70.2 | 71.7 | 71.6 | 69.8 | 71.2 | 71.0 | 72.5 | 68.8 | 66.8 | 68.9 | 70.7 | 68.3 | 70.9 | 69.0 | 71.2 | 68.9 | 72.6 | 70.6 | 68.7 |
| | gemini-2.0-flash | 79.4 | 76.0 | 77.2 | 77.1 | 78.9 | 76.0 | 76.0 | 78.8 | 74.2 | 73.6 | 75.0 | 75.0 | 74.3 | 77.1 | 74.3 | 77.1 | 75.5 | 78.5 | 77.0 | 71.7 |
| | gemini-2.5-pro | 81.0 | 79.4 | 81.3 | 79.6 | 80.8 | 79.1 | 79.2 | 80.7 | 77.7 | 77.8 | 78.9 | 79.3 | 75.4 | 81.3 | 78.8 | 80.0 | 77.5 | 82.6 | 81.2 | 77.8 |
| | gpt-4o-mini | 69.4 | 68.9 | 70.1 | 70.7 | 70.0 | 68.8 | 68.4 | 69.8 | 68.1 | 67.1 | 68.8 | 68.8 | 68.8 | 69.8 | 67.7 | 68.7 | 68.1 | 70.5 | 69.5 | 66.3 |
| | o4-mini | 76.1 | 73.4 | 73.7 | 72.9 | 75.9 | 73.7 | 73.1 | 74.7 | 72.9 | 72.1 | 71.7 | 74.2 | 72.8 | 72.9 | 71.4 | 75.3 | 71.0 | 76.1 | 74.3 | 72.7 |
| GSM8K | Qwen2.5-72B | 94.9 | 93.1 | 94.3 | 93.4 | 93.9 | 93.6 | 92.5 | 93.9 | 91.7 | 93.5 | 94.3 | 88.6 | 94.2 | 91.8 | 94.7 | 91.2 | 94.8 | 94.5 | 90.8 |
| | DeepSeek-R1-70B | 91.1 | 89.7 | 90.8 | 90.6 | 92.0 | 89.2 | 89.2 | 91.2 | 88.2 | 88.9 | 91.3 | 87.9 | 85.7 | 91.0 | 87.0 | 91.4 | 88.2 | 91.5 | 92.0 | 88.0 |
| | Llama-3.3-70B | 96.1 | 94.1 | 95.2 | 94.2 | 94.7 | 93.9 | 93.7 | 95.5 | 93.1 | 93.6 | 95.7 | 94.1 | 90.3 | 95.5 | 92.0 | 95.2 | 92.7 | 96.3 | 95.9 | 91.3 |
| | gemini-2.0-flash | 95.1 | 93.3 | 94.8 | 93.9 | 94.7 | 93.4 | 92.8 | 95.5 | 90.4 | 93.2 | 94.7 | 93.3 | 88.3 | 94.6 | 91.4 | 94.5 | 91.4 | 95.4 | 95.1 | 91.3 |
| | gemini-2.5-pro | 96.1 | 94.0 | 95.0 | 94.0 | 95.6 | 93.5 | 93.5 | 95.8 | 91.7 | 94.2 | 95.2 | 93.3 | 91.4 | 95.7 | 91.7 | 95.4 | 92.0 | 95.9 | 95.6 | 91.7 |
| | gpt-4o-mini | 93.0 | 90.5 | 92.0 | 91.3 | 93.0 | 90.8 | 90.1 | 91.6 | 87.9 | 89.2 | 92.3 | 90.1 | 84.8 | 92.2 | 88.8 | 92.0 | 88.6 | 92.1 | 93.3 | 88.5 |
| | o4-mini | 96.4 | 94.2 | 95.7 | 94.9 | 95.1 | 94.2 | 94.2 | 95.5 | 92.6 | 93.5 | 95.8 | 94.2 | 88.0 | 96.2 | 91.7 | 96.1 | 92.9 | 96.1 | 95.8 | 93.1 |
| HellaSwag | Qwen2.5-72B | 90.1 | 87.4 | 88.0 | 86.4 | 88.9 | 87.9 | 87.4 | 88.4 | 86.8 | 85.4 | 87.0 | 88.1 | 85.6 | 88.6 | 85.8 | 88.7 | 86.9 | 89.2 | 88.8 | 85.9 |
| | DeepSeek-R1-70B | 83.3 | 82.1 | 81.8 | 81.2 | 82.8 | 82.3 | 82.0 | 82.9 | 81.9 | 80.6 | 82.3 | 82.1 | 80.8 | 82.7 | 81.4 | 82.8 | 81.8 | 83.2 | 82.9 | 81.5 |
| | Llama-3.3-70B | 88.5 | 87.0 | 87.2 | 86.0 | 88.4 | 87.4 | 87.1 | 87.7 | 86.5 | 85.8 | 86.8 | 87.3 | 85.0 | 87.7 | 86.2 | 87.9 | 86.6 | 88.0 | 88.1 | 86.2 |
| | gemini-2.0-flash | 90.2 | 87.8 | 88.5 | 87.4 | 89.5 | 88.6 | 87.8 | 88.7 | 86.5 | 85.9 | 87.7 | 88.0 | 85.1 | 89.0 | 86.9 | 89.3 | 87.4 | 89.4 | 89.4 | 86.1 |
| | gemini-2.5-flash | 92.1 | 90.3 | 90.3 | 90.0 | 91.5 | 91.0 | 90.3 | 91.2 | 89.9 | 88.9 | 89.9 | 90.6 | 87.5 | 91.3 | 89.2 | 91.5 | 89.9 | 91.5 | 91.1 | 90.0 |
| | gpt-4o-mini | 86.0 | 84.0 | 84.1 | 83.3 | 85.2 | 84.5 | 84.3 | 84.4 | 83.4 | 82.7 | 84.0 | 84.5 | 82.4 | 85.1 | 83.1 | 84.9 | 83.7 | 85.5 | 85.5 | 82.3 |
| | o4-mini | 79.4 | 86.0 | 86.0 | 85.6 | 87.3 | 86.4 | 86.1 | 86.8 | 84.8 | 84.6 | 85.4 | 86.4 | 84.3 | 86.8 | 85.2 | 86.8 | 86.0 | 86.8 | 86.6 | 85.4 |
| WinoGrande | Qwen2.5-72B | 83.7 | 71.5 | 73.8 | 72.8 | 76.9 | 76.0 | 70.4 | 77.0 | 67.7 | 63.2 | 64.9 | 71.3 | 62.8 | 75.5 | 67.0 | 74.4 | 69.1 | 82.0 | 74.7 | 66.6 |
| | DeepSeek-R1-70B | 86.4 | 74.0 | 76.3 | 73.3 | 78.8 | 79.3 | 74.5 | 79.9 | 71.3 | 65.0 | 69.4 | 75.5 | 67.5 | 78.3 | 68.8 | 76.2 | 69.1 | 85.5 | 76.4 | 66.3 |
| | Llama-3.3-70B | 82.3 | 71.6 | 76.0 | 70.9 | 75.5 | 74.3 | 71.0 | 78.1 | 68.0 | 63.8 | 67.4 | 71.4 | 63.1 | 75.5 | 68.0 | 73.6 | 68.6 | 81.5 | 75.0 | 67.5 |
| | gemini-2.0-flash | 73.2 | 64.4 | 68.1 | 64.3 | 70.2 | 67.0 | 65.4 | 68.4 | 62.0 | 60.4 | 60.7 | 63.8 | 57.5 | 68.5 | 60.5 | 65.3 | 59.1 | 71.5 | 65.8 | 60.8 |
| | gemini-2.5-pro | 91.1 | 79.4 | 81.0 | 79.8 | 86.4 | 84.5 | 78.3 | 86.4 | 77.1 | 70.6 | 73.6 | 82.6 | 68.3 | 83.3 | 73.9 | 82.4 | 77.1 | 90.0 | 81.1 | 71.8 |
| | gpt-4o-mini | 70.5 | 60.6 | 61.8 | 60.9 | 66.1 | 61.5 | 60.1 | 64.2 | 57.9 | 57.7 | 57.3 | 60.6 | 54.5 | 62.6 | 58.7 | 61.9 | 58.5 | 68.0 | 60.9 | 57.3 |
| | o4-mini | 85.6 | 74.9 | 78.7 | 74.0 | 80.7 | 79.7 | 73.0 | 81.8 | 70.8 | 64.8 | 70.2 | 75.8 | 66.7 | 79.1 | 69.9 | 80.0 | 72.3 | 85.4 | 76.1 | 68.9 |

majority approval from the annotators. All models, except LLaMA-3.1-8B-Instruct, achieved over 90% validity, indicating that our framework is broadly compatible with high-capacity LLMs.

We also evaluated the semantic checker model $S$ using the outputs from LLaMA-3.3-70B-Instruct [26]. Specifically, we randomly sampled 200 outputs that $S$ rejected and 200 that it accepted. Human annotators then labeled whether each output preserved the original semantics. Using these human annotations as gold labels, $S$ achieved a precision of 83.6%, recall of 97.0%, and F1 score of 89.8%. These results indicate that $S$ performs reliably in distinguishing meaning-preserving transformations. Additional details on the human evaluation procedure are provided in Appendix C.7.

# 4 Experiments

We evaluated the transformed datasets on seven state-of-the-art models: Qwen2.5-72B-Instruct [67], DeepSeek-R1-Distill-Llama-70B [14], LLaMA-3.3-70B-Instruct [26], Gemini 2.0 Flash [23], Gemini 2.5 Pro [22], GPT-4o-mini [50], and o4-mini [52].[13]

## 4.1 Experiment Results

Tables 4 and 5 report model performances on dialect and ESL English varieties, respectively. "Orig." denotes the original SAE dataset, and "Mean" represents the average performance across all dialects or varieties of a CEFR level, excluding the SAE dataset. Values in blue are those that performed better than the original. Cell color reflects performance deviation across varieties.

Overall, models tend to perform worse on non-SAE varieties, with maximum performance drops of 12.5% for dialects and 46.3% for ESL English. Models with strong reasoning capabilities, such as DeepSeek-R1-70B or o4-mini, exhibit greater robustness, suggesting that reasoning ability may

---

[13]Model versions: `gemini-2.5-pro-exp-03-25`, `gpt-4o-mini-2024-07-18`, `o4-mini-2025-04-16`

Table 5: Benchmark results of seven models on ESL English varieties. Values in blue cells are those that performed better than the original.

| | | Orig. | CEFR A | | | | | | | | | | | CEFR B | | | | | | | | | | |
|---|---|---|---|---|---|---|---|---|---|---|---|---|---|---|---|---|---|---|---|---|---|---|---|---|
| | | | Mean | ar | zh | fr | de | it | ja | pt | ru | es | tr | Mean | ar | zh | fr | de | it | ja | pt | ru | es | tr |
| **MMLU** | Qwen2.5-72B | 84.6 | 66.6 | 63.3 | 67.4 | 71.1 | 66.9 | 72.2 | 66.7 | 65.0 | 65.4 | 63.8 | 64.0 | 69.1 | 65.7 | 70.2 | 73.4 | 69.1 | 74.2 | 68.7 | 67.5 | 69.2 | 65.9 | 67.4 |
| | DeepSeek-R1-70B | 84.3 | 71.5 | 69.3 | 71.7 | 74.2 | 71.8 | 75.3 | 70.7 | 70.2 | 71.1 | 71.1 | 70.1 | 73.8 | 70.4 | 74.9 | 75.7 | 73.2 | 77.5 | 74.3 | 73.1 | 73.8 | 72.2 | 72.9 |
| | Llama-3.3-70B | 84.3 | 71.3 | 68.8 | 71.5 | 74.9 | 71.9 | 75.9 | 70.9 | 69.9 | 70.5 | 69.3 | 69.3 | 73.4 | 71.2 | 74.2 | 76.6 | 73.2 | 77.0 | 73.3 | 72.0 | 73.5 | 71.2 | 72.1 |
| | gemini-2.0-flash | 86.0 | 67.2 | 64.2 | 68.4 | 71.4 | 67.5 | 71.8 | 67.5 | 65.8 | 66.2 | 64.7 | 64.7 | 70.6 | 67.4 | 71.8 | 74.5 | 70.4 | 75.6 | 70.0 | 69.1 | 70.5 | 67.8 | 68.5 |
| | gemini-2.5-pro | 90.9 | 76.0 | 74.3 | 76.4 | 79.2 | 76.4 | 80.0 | 76.2 | 74.9 | 74.9 | 74.0 | 73.5 | 77.1 | 74.4 | 77.8 | 80.1 | 77.1 | 80.9 | 76.6 | 76.1 | 77.3 | 75.1 | 76.1 |
| | gpt-4o-mini | 76.3 | 59.0 | 56.0 | 63.7 | 62.2 | 58.9 | 62.6 | 57.9 | 57.6 | 58.1 | 56.4 | 56.5 | 63.7 | 61.1 | 65.0 | 67.1 | 63.5 | 68.5 | 63.2 | 62.3 | 63.5 | 61.4 | 61.8 |
| | o4-mini | 89.3 | 73.8 | 71.3 | 74.6 | 78.0 | 74.0 | 78.2 | 73.9 | 72.6 | 72.6 | 71.8 | 71.3 | 74.9 | 71.8 | 76.1 | 78.8 | 74.6 | 79.7 | 74.4 | 73.5 | 74.7 | 72.4 | 73.3 |
| **ARC** | Qwen2.5-72B | 96.0 | 83.4 | 78.4 | 86.0 | 90.1 | 83.3 | 91.7 | 82.6 | 79.6 | 81.7 | 80.1 | 80.7 | 82.4 | 76.9 | 82.1 | 89.5 | 82.9 | 91.7 | 82.9 | 77.9 | 82.2 | 77.7 | 79.9 |
| | DeepSeek-R1-70B | 95.3 | 86.8 | 83.1 | 87.9 | 91.5 | 86.8 | 92.9 | 85.8 | 84.2 | 85.4 | 86.0 | 84.4 | 85.9 | 82.6 | 85.4 | 91.2 | 86.7 | 92.6 | 85.7 | 82.9 | 86.7 | 82.0 | 83.6 |
| | Llama-3.3-70B | 95.2 | 85.3 | 81.4 | 87.7 | 90.4 | 85.5 | 92.4 | 84.1 | 83.2 | 82.9 | 83.2 | 82.4 | 85.5 | 81.7 | 85.3 | 90.4 | 86.0 | 92.2 | 86.5 | 82.3 | 85.3 | 82.1 | 83.6 |
| | gemini-2.0-flash | 95.9 | 83.3 | 78.7 | 85.5 | 89.4 | 84.2 | 91.3 | 81.9 | 79.2 | 82.4 | 80.5 | 80.1 | 83.3 | 78.0 | 83.0 | 90.4 | 83.9 | 92.0 | 82.9 | 80.0 | 83.1 | 79.6 | 80.6 |
| | gemini-2.5-pro | 97.5 | 87.8 | 83.3 | 89.4 | 93.4 | 88.8 | 94.1 | 86.8 | 85.3 | 86.4 | 85.5 | 85.3 | 87.1 | 83.1 | 87.1 | 92.9 | 87.4 | 93.9 | 87.1 | 84.3 | 87.1 | 83.4 | 85.1 |
| | gpt-4o-mini | 92.2 | 78.4 | 74.2 | 81.3 | 84.9 | 79.1 | 86.0 | 77.8 | 74.9 | 75.6 | 75.7 | 74.8 | 79.2 | 73.1 | 79.1 | 86.3 | 80.7 | 87.9 | 79.2 | 75.1 | 78.7 | 75.0 | 76.8 |
| | o4-mini | 97.0 | 86.4 | 81.9 | 87.0 | 92.8 | 87.9 | 93.4 | 85.5 | 83.1 | 85.3 | 84.4 | 83.1 | 85.3 | 80.3 | 84.8 | 90.9 | 85.3 | 92.9 | 86.7 | 82.1 | 84.7 | 82.3 | 82.6 |
| **TruthfulQA** | Qwen2.5-72B | 77.0 | 69.6 | 67.7 | 71.7 | 71.7 | 68.1 | 73.5 | 68.1 | 67.3 | 69.2 | 69.2 | 69.3 | 70.2 | 65.3 | 70.7 | 73.2 | 70.7 | 72.5 | 69.0 | 70.4 | 70.6 | 70.7 | 68.8 |
| | DeepSeek-R1-70B | 72.0 | 66.9 | 66.9 | 66.6 | 66.1 | 65.5 | 67.3 | 67.9 | 66.6 | 69.0 | 66.0 | 67.1 | 67.3 | 64.9 | 66.1 | 70.2 | 66.8 | 66.7 | 67.1 | 67.5 | 69.1 | 67.7 | 67.3 |
| | Llama-3.3-70B | 71.2 | 63.7 | 63.2 | 65.3 | 64.4 | 62.9 | 65.3 | 63.6 | 63.4 | 62.4 | 64.2 | 62.6 | 64.6 | 63.0 | 64.8 | 65.9 | 64.4 | 66.8 | 63.6 | 65.8 | 64.3 | 63.6 | 63.5 |
| | gemini-2.0-flash | 79.4 | 68.9 | 66.9 | 69.5 | 69.3 | 68.9 | 72.1 | 69.3 | 67.7 | 69.5 | 67.1 | 68.5 | 69.0 | 66.2 | 69.0 | 70.7 | 69.0 | 72.2 | 68.2 | 68.0 | 70.0 | 67.9 | 69.1 |
| | gemini-2.5-pro | 81.0 | 73.5 | 72.7 | 74.8 | 73.7 | 72.9 | 75.3 | 73.0 | 74.2 | 72.2 | 72.2 | 73.8 | 75.0 | 73.2 | 75.0 | 77.0 | 74.4 | 75.0 | 75.8 | 74.5 | 77.0 | 74.1 | 74.1 |
| | gpt-4o-mini | 69.4 | 62.9 | 60.8 | 63.6 | 63.2 | 63.2 | 66.8 | 62.6 | 62.1 | 63.6 | 61.8 | 61.2 | 63.5 | 60.4 | 63.8 | 64.8 | 65.6 | 65.6 | 62.0 | 63.4 | 65.4 | 61.7 | 62.1 |
| | o4-mini | 76.1 | 66.8 | 65.2 | 67.1 | 67.3 | 66.9 | 69.7 | 67.3 | 65.7 | 66.6 | 66.3 | 66.3 | 68.0 | 64.1 | 68.9 | 70.0 | 68.8 | 69.4 | 67.2 | 68.2 | 69.4 | 66.8 | 66.7 |
| **GSM8K** | Qwen2.5-72B | 94.9 | 75.3 | 68.6 | 72.9 | 85.0 | 78.4 | 86.2 | 78.3 | 70.4 | 74.0 | 69.7 | 69.9 | 78.0 | 71.2 | 75.5 | 89.4 | 83.2 | 90.3 | 82.2 | 70.7 | 75.4 | 71.6 | 70.6 |
| | DeepSeek-R1-70B | 91.1 | 75.4 | 69.2 | 73.5 | 85.3 | 78.2 | 85.3 | 79.5 | 70.0 | 72.6 | 70.1 | 69.8 | 78.2 | 70.7 | 75.6 | 89.6 | 82.8 | 90.7 | 83.0 | 71.6 | 76.0 | 71.1 | 70.4 |
| | Llama-3.3-70B | 96.1 | 76.5 | 70.2 | 75.1 | 86.1 | 79.6 | 87.9 | 79.9 | 70.1 | 73.7 | 71.9 | 71.6 | 79.2 | 73.0 | 77.0 | 91.3 | 83.5 | 91.9 | 83.2 | 71.9 | 76.0 | 72.1 | 71.6 |
| | gemini-2.0-flash | 95.1 | 74.5 | 67.8 | 71.9 | 84.2 | 78.2 | 85.8 | 76.9 | 68.7 | 72.0 | 69.8 | 69.1 | 78.1 | 72.0 | 76.0 | 88.7 | 83.6 | 91.2 | 81.7 | 71.0 | 75.2 | 71.3 | 70.0 |
| | gemini-2.5-pro | 96.1 | 76.5 | 70.2 | 75.3 | 86.1 | 79.9 | 86.6 | 80.0 | 70.4 | 74.7 | 70.8 | 71.0 | 79.2 | 72.5 | 76.7 | 91.5 | 83.7 | 91.1 | 82.7 | 72.4 | 76.7 | 72.3 | 72.1 |
| | gpt-4o-mini | 93.0 | 71.6 | 64.9 | 70.5 | 81.4 | 74.7 | 82.9 | 74.6 | 66.0 | 67.9 | 66.9 | 66.4 | 75.0 | 68.1 | 72.7 | 86.6 | 79.8 | 87.7 | 78.5 | 67.3 | 73.3 | 68.3 | 67.4 |
| | o4-mini | 96.4 | 76.3 | 70.8 | 73.5 | 86.3 | 79.8 | 87.7 | 79.5 | 70.1 | 73.4 | 71.0 | 70.8 | 79.5 | 73.1 | 77.0 | 91.0 | 84.6 | 92.6 | 82.7 | 72.3 | 77.1 | 72.3 | 72.0 |
| **HellaSwag** | Qwen2.5-72B | 90.1 | 76.6 | 73.6 | 78.0 | 80.6 | 76.4 | 80.9 | 75.7 | 75.5 | 77.0 | 74.4 | 74.1 | 78.2 | 74.9 | 79.8 | 83.9 | 78.6 | 83.7 | 76.8 | 76.8 | 78.2 | 75.6 | 73.9 |
| | DeepSeek-R1-70B | 83.3 | 75.7 | 74.2 | 77.0 | 77.6 | 74.9 | 78.0 | 75.3 | 75.3 | 76.0 | 74.1 | 74.9 | 76.6 | 75.0 | 77.4 | 79.6 | 76.7 | 80.0 | 75.9 | 76.1 | 76.3 | 74.9 | 74.7 |
| | Llama-3.3-70B | 88.5 | 79.6 | 78.0 | 80.5 | 82.1 | 79.2 | 82.1 | 78.8 | 79.3 | 79.3 | 78.4 | 78.3 | 81.6 | 80.1 | 82.7 | 85.2 | 81.2 | 85.0 | 80.5 | 80.7 | 81.3 | 80.2 | 79.5 |
| | gemini-2.0-flash | 90.2 | 78.3 | 76.3 | 79.4 | 81.1 | 77.9 | 81.5 | 77.2 | 77.8 | 78.2 | 77.0 | 77.2 | 81.1 | 78.9 | 82.4 | 84.8 | 81.0 | 85.1 | 79.8 | 80.4 | 80.9 | 78.9 | 78.4 |
| | gemini-2.5-pro | 92.1 | 83.6 | 82.6 | 85.1 | 85.5 | 83.2 | 86.4 | 82.7 | 82.7 | 83.4 | 82.1 | 82.1 | 85.2 | 83.5 | 86.2 | 88.2 | 85.3 | 88.8 | 83.9 | 84.7 | 85.2 | 83.6 | 82.9 |
| | gpt-4o-mini | 86.0 | 74.6 | 72.4 | 76.3 | 78.0 | 74.6 | 78.5 | 73.3 | 73.1 | 74.0 | 72.6 | 73.0 | 76.4 | 74.2 | 77.5 | 81.3 | 76.4 | 81.0 | 74.6 | 75.5 | 76.0 | 74.2 | 73.5 |
| | o4-mini | 79.4 | 77.9 | 76.5 | 79.1 | 80.8 | 77.7 | 81.2 | 76.6 | 77.0 | 78.1 | 76.2 | 76.2 | 78.2 | 80.0 | 83.4 | 78.5 | 83.3 | 77.6 | 78.4 | 78.7 | 76.5 | 75.9 | |
| **WinoGrande** | Qwen2.5-72B | 83.7 | 39.7 | 36.0 | 40.6 | 46.5 | 37.4 | 45.5 | 38.4 | 39.4 | 38.4 | 37.4 | 37.6 | 36.8 | 33.1 | 40.7 | 42.6 | 36.4 | 42.9 | 33.7 | 33.9 | 38.4 | 31.6 | 34.0 |
| | DeepSeek-R1-70B | 86.4 | 46.8 | 47.4 | 46.8 | 46.7 | 47.4 | 48.0 | 46.0 | 49.1 | 44.4 | 45.4 | 46.7 | 45.7 | 42.6 | 47.0 | 46.6 | 46.0 | 47.4 | 46.7 | 44.9 | 47.8 | 44.9 | 43.1 |
| | Llama-3.3-70B | 82.3 | 47.4 | 46.1 | 47.0 | 48.6 | 47.1 | 49.9 | 47.2 | 47.2 | 46.1 | 47.1 | 47.4 | 46.9 | 45.1 | 49.1 | 48.8 | 44.9 | 48.8 | 46.7 | 45.8 | 46.2 | 45.3 | |
| | gemini-2.0-flash | 73.2 | 40.4 | 35.9 | 41.3 | 42.2 | 37.6 | 48.7 | 39.3 | 39.3 | 39.5 | 41.2 | 38.9 | 40.1 | 37.6 | 43.7 | 45.8 | 38.1 | 47.6 | 36.4 | 36.5 | 41.2 | 35.6 | 38.4 |
| | gemini-2.5-pro | 91.1 | 44.8 | 43.9 | 45.1 | 44.9 | 44.7 | 47.8 | 44.0 | 44.3 | 42.9 | 45.0 | 45.4 | 44.0 | 41.9 | 45.1 | 46.5 | 42.7 | 48.2 | 41.6 | 44.2 | 45.6 | 41.9 | 42.5 |
| | gpt-4o-mini | 70.5 | 38.5 | 33.2 | 39.4 | 42.8 | 37.0 | 45.2 | 37.2 | 37.8 | 39.0 | 38.4 | 35.3 | 37.0 | 33.3 | 41.3 | 43.5 | 37.0 | 41.2 | 34.3 | 35.9 | 38.1 | 32.8 | 33.1 |
| | o4-mini | 85.6 | 44.0 | 43.6 | 44.8 | 45.7 | 41.8 | 47.2 | 43.6 | 43.6 | 43.3 | 42.6 | 43.9 | 42.5 | 39.9 | 44.7 | 44.1 | 41.9 | 46.7 | 40.1 | 42.3 | 45.6 | 40.1 | 40.1 |

support robustness in varieties. Also, performance degradation is more pronounced in reasoning-based QA (4.7% for dialects and 22.6% for ESL English) compared to knowledge-based QA datasets (1.3% for dialects and 10.9% for ESL English) on average, implying that reasoning tasks are more sensitive to linguistic variations. Additionally, the overall performance drop is bigger for ESL English than for dialects. In 4 out of the 6 datasets, CEFR A varieties yield lower mean scores than those of CEFR B, indicating that simpler or a higher deviance SAE pose greater challenges for LLMs.

Certain English varieties exhibit consistent relative performance patterns across datasets and models. Among dialects, Newfoundland English (NFE), Welsh English (WeE), and Irish English (IrE) tend to yield lower scores, whereas Australian English (AuE), Southeast English (SE-Eng), and Southwest English (SW-Eng) show relatively stronger performance. For ESL varieties, Arabic (ar) and Turkish (tr) underperform, while French (fr) and Italian (it) achieve higher scores. We attribute these trends partly to data availability. Dialects with lower performance tend to have fewer native speakers (on the order of millions), while better-performing dialects are spoken in regions where English is an official language. Similarly, Arabic and Turkish are under-represented in two multilingual pretraining corpora (1.66% and 1.93% in mC4 [71]; 2.0% and 1.7% in OSCAR [53], respectively) while French and Italian are better represented (2.89% and 2.43% in mC4; 8.9% and 5.3% in OSCAR, respectively). These observations suggest that the amount of pretraining data available for each variety is a critical factor influencing the LLM robustness across varieties.

## 4.2 Performance Gap Analysis with Sentence Length

We have separated the six datasets based on word count and sentence count using four quantiles, and we report the average performance differences across models accordingly. All datasets except for GSM8K [12] exhibit a linear trend in which performance degrades as sentence length increases, whether measured by word count or sentence count. We hypothesize that GSM8K [12] does not

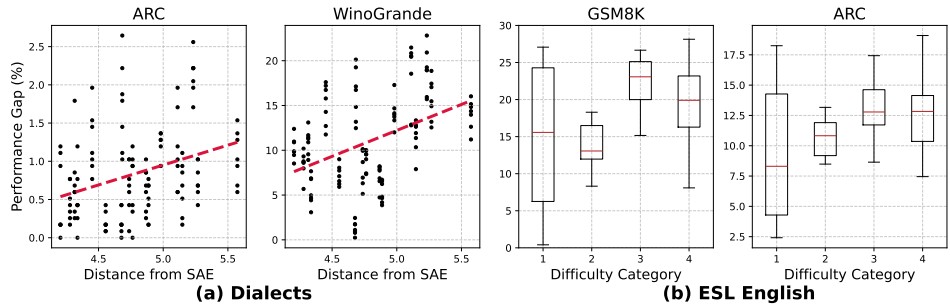

Figure 2: Correlation between linguistic distance and model performance degradation.

follow this trend because, unlike the other five datasets, its question lengths are relatively uniform. As a result, there are no substantial length differences across quantiles.

## 4.3 Correlation between Linguistic Distance with Performance Degradation

We investigate the relationship between LLM performance and the linguistic distance from SAE. We represent each dialect variety as a 235-dimensional vector using the features from the eWAVE database and encoding its prevalence as values. We apply singular value decomposition to reduce dimensionality while preserving

| Datset | Word-Level | | | | Sentence-Level | | | |
|---|---|---|---|---|---|---|---|---|
| | Q1 | Q2 | Q3 | Q4 | Q1 | Q2 | Q3 | Q4 |
| MMLU | 0.007 | 0.014 | 0.021 | 0.029 | 0.013 | 0.017 | 0.017 | 0.029 |
| ARC | 0.009 | 0.007 | 0.010 | 0.012 | 0.006 | 0.011 | 0.014 | 0.010 |
| TruthfulQA | 0.014 | 0.017 | 0.013 | 0.020 | 0.010 | 0.011 | 0.013 | 0.054 |
| GSM8K | 0.026 | 0.028 | 0.018 | 0.014 | 0.027 | 0.018 | 0.027 | 0.012 |
| HellaSwag | 0.022 | 0.023 | 0.027 | 0.032 | 0.022 | 0.026 | 0.032 | 0.034 |
| WinoGrande | 0.104 | 0.114 | 0.119 | 0.125 | 0.114 | 0.11 | 0.112 | 0.103 |

over 90% of the variance, and compute the Euclidean distance between each dialect and the SAE reference.[14] Figure 2 (a) plots the linguistic distance against performance degradation in ARC and WinoGrande, revealing a positive correlation. This result indicates that models perform worse on dialects that are linguistically farther from SAE.

A similar pattern was observed in ESL English. The Defense Language Institute Foreign Language Center (DLIFLC) categorizes non-English languages into four levels of difficulty for native English speakers. Category 1 (easiest) includes French, Italian, Portuguese, and Spanish; Category 2 includes German; Category 3 includes Russian and Turkish; and Category 4 (hardest) includes Arabic, Chinese-Mandarin, and Japanese. Figure 2 (b) presents performance degradation by L1 category using box plots in GSM8K and ARC. The results show that, although there is variance within each category, ESL English derived from categories 1 and 2 yield smaller performance drops, while those from categories 3 and 4 lead to more significant degradation. These findings indicate that LLMs are strongly biased toward SAE, and that their robustness declines as the linguistic properties of the target variety diverge from it. Plots and correlation values for all datasets are presented in Appendix D.2 and D.3.

## 4.4 Experiments on Open-Ended Tasks

We conducted experiments using three open-ended evaluation setups: IFEval [76], AlpacaFarm [16], and MT-Bench [75], selecting 8 dialects and 10 ESL variants. We evaluated four models: Qwen2.5-72B-Instruct [67], DeepSeek-R1-Distill-Llama-70B [14], LLaMA-3.3-70B-Instruct [26], and o4-mini [52]. We used LLM-as-a-judge strategy, using GPT-4.1[15] [1] as the evaluator. We translated the original instructions into the respective English variants and generated model outputs based on these transformed instructions. For IFEval, the metric is the accuracy of instruction adherence. For AlpacaFarm, we report the win rate when comparing model outputs against GPT-4. For MT-Bench, the evaluation LLM provides a score from 0 to 10 indicating how well the model's output aligns with the given instruction. Table 6 reports the model performances on selected dialect and ESL English varieties. Consistent with the results observed in closed-form tasks, model performance declines across all three datasets when evaluated on the transformed instructions. Notably, Irish English, Newfoundland English, and Welsh English exhibited lower performance compared to African

---

[14]We used Colloquial American English as SAE reference.
[15]Model version: `gpt-4.1-2025-04-14`

Table 6: Open-ended task benchmark results of four models on selected dialect and ESL varieties. Values in blue cells are those that performed better than the original.

| | | Orig. | AAVE | AuE | TdCE | IrE | NFE | WeE | SE-Eng | NE-Eng | ar | it | es | fr | tr | ar | it | es | fr | tr |
|---|---|---|---|---|---|---|---|---|---|---|---|---|---|---|---|---|---|---|---|---|
| | | | | | | Dialect | | | | | ESL (CEFR A) | | | | | ESL (CEFR B) | | | | |
| IFEval | Qwen-2.5-72B | 82.4 | 78.7 | 75.2 | 77.6 | 69.5 | 70.4 | 72.1 | 80.2 | 75.8 | 60.7 | 62.8 | 62.8 | 63.5 | 61 | 72.5 | 74.3 | 72.3 | 73.3 | 71 |
| | DeepSeek-R1-70B | 76.5 | 72.6 | 71 | 72.3 | 64.1 | 65.2 | 66.5 | 71.3 | 69.9 | 59.7 | 57.4 | 59.2 | 55.9 | 56.4 | 65.6 | 66.4 | 64.1 | 62.4 | 64.2 |
| | LLaMA-3.3-70B | 88.4 | 83.7 | 82.4 | 84.5 | 75.8 | 75.8 | 77.6 | 83 | 81.3 | 66.8 | 66.6 | 67.6 | 67.1 | 66.8 | 77 | 77.9 | 77 | 75.8 | 76.8 |
| | o4-mini | 91.5 | 74.9 | 74.3 | 75.6 | 65.8 | 68.4 | 67.1 | 73.6 | 72.1 | 57.7 | 59.2 | 57.9 | 59.9 | 58.9 | 67.1 | 69.1 | 67.8 | 68.2 | 66.7 |
| AlpacaFarm | Qwen-2.5-72B | 59.0 | 57.7 | 57.7 | 59.1 | 57.4 | 59.6 | 58.6 | 57.9 | 54.7 | 53.9 | 54.5 | 55.3 | 57.1 | 57.0 | 54.3 | 56.8 | 58.0 | 56.8 | 58.5 |
| | DeepSeek-R1-70B | 46.8 | 45.3 | 45.3 | 43.0 | 45.5 | 44.6 | 43.7 | 42.1 | 47.6 | 41.9 | 38.3 | 43.2 | 43.5 | 38.6 | 43.9 | 44.2 | 43.9 | 42.0 | 40.3 |
| | LLaMA-3.3-70B | 39.8 | 38.6 | 36.6 | 36.9 | 37.7 | 36.9 | 36.6 | 38.2 | 36.6 | 32.6 | 34.9 | 36.1 | 36.0 | 36.0 | 34.9 | 36.9 | 37.7 | 37.2 | 36.7 |
| | o4-mini | 84.0 | 40.9 | 39.8 | 41.2 | 40.8 | 41.1 | 40.8 | 40.2 | 41.3 | 76.7 | 77.6 | 78.8 | 81.3 | 81.1 | 77.4 | 80.9 | 82.6 | 80.8 | 83.3 |
| MT-Bench | Qwen-2.5-72B | 9.0 | 8.9 | 9.1 | 8.9 | 8.6 | 8.5 | 8.9 | 9.0 | 9.0 | 8.0 | 7.3 | 7.4 | 7.4 | 7.5 | 8.7 | 8.2 | 8.2 | 8.0 | 8.3 |
| | DeepSeek-R1-70B | 8.6 | 8.7 | 8.9 | 8.5 | 8.3 | 8.2 | 8.8 | 8.6 | 8.6 | 7.9 | 7.2 | 7.1 | 7.3 | 7.1 | 8.2 | 7.8 | 7.3 | 7.6 | 7.4 |
| | LLaMA-3.3-70B | 9.1 | 8.5 | 9.2 | 8.6 | 8.7 | 8.3 | 8.7 | 8.9 | 9.0 | 8.2 | 7.2 | 7.3 | 7.3 | 7.2 | 8.7 | 8.1 | 8.2 | 8.4 | 8.3 |
| | o4-mini | 9.4 | 8.6 | 9.0 | 8.6 | 8.6 | 8.3 | 8.9 | 9.1 | 9.0 | 8.1 | 7.5 | 7.2 | 7.4 | 7.5 | 8.5 | 8.4 | 8.4 | 8.2 | 8.2 |

American Vernacular English, Australian English, and Tristan da Cunha English, mirroring trends observed in the closed-form evaluations. Similarly, ESL variants at CEFR level B outperformed those at level A, reflecting a pattern consistent with earlier findings.

# 5 Conclusion

In this paper, we introduce Trans-EnV, a framework that automatically transforms SAE datasets into a wide range of target English varieties. Trans-EnV is grounded in expert-curated linguistic resources, validated through rigorous experimentation, and developed with the guidance of a researcher specializing in second language acquisition. By leveraging an LLM for transformations, our framework is scalable across various datasets and varieties. When applied to benchmark datasets, Trans-EnV achieves high transformation coverage, over 80%, and human evaluations confirmed its linguistic validity. We transform six benchmark datasets into 38 variates, and experimental results with seven state-of-the-art LLMs reveal significant performance degradation on non-standard varieties, underscoring the importance of evaluating linguistic robustness across diverse forms of English.

**Limitations & Future Work.** This work focuses primarily on English varieties. Extending Trans-EnV to other languages would require language-specific resources and transformation guidelines, which we leave for future work. Our evaluation is currently limited to QA tasks as an initial step toward assessing the linguistic robustness. Extending to other tasks remains as future research.

**Broader Impacts.** This work aims to improve the robustness of LLMs for non-standard English speaking users. Trans-EnV supports the broader goals of global accessibility and social responsibility in language technologies. Nonetheless, the absence of professional linguistic resources may lead to invalid transformations. To mitigate this risk, we encourage researchers to incorporate expert-curated linguistic datasets when adapting or extending Trans-EnV.

# Acknowledgment

This work was supported by the Institute of Information & Communications Technology Planning & Evaluation (IITP) grants (No.RS-2019-II190075, No.RS-2022-II220984), the Korea Health Industry Development Institute (KHIDI) grant (No.RS-2025-02213750) and the National Research Foundation of Korea (NRF) grants (NRF-2020H1D3A2A03100945), funded by the Korea government (MSIT, MOHW).

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

# Appendix

# A Datasheet for Datasets

The following section is answers to questions listed in datasheets for datasets.

## A.1 Motivation

- Question: **For what purpose was the dataset created?** Was there a specific task in mind? Was there a specific gap that needed to be filled? Please provide a description.

  Answer: To evaluate the linguistic robustness of language models across diverse English varieties by transforming Standard American English (SAE) datasets.

- Question: **Who created the dataset (e.g., which team, research group) and on behalf of which entity (e.g., company, institution, organization)?**

  Answer: The authors of this paper.

- Question: **Who funded the creation of the dataset?** If there is an associated grant, please provide the name of the grantor and the grant name and number.

  Answer: This work was supported by Institute for Information & communications Technology Planning & Evaluation(IITP) grant funded by the Korea government(MSIT) (RS-2019-II190075, Artificial Intelligence Graduate School Program(KAIST)).

## A.2 Composition

- Question: **What do the instances that comprise the dataset represent (e.g., documents, photos, people, countries)?** Are there multiple types of instances (e.g., movies, users, and ratings; people and interactions between them; nodes and edges)? Please provide a description.

  Answer: QA datasets (sentences) transformed into various English varieties.

- Question: **How many instances are there in total (of each type, if appropriate)?**

  Answer: There are about 952K instances in total.

- Question: **Does the dataset contain all possible instances or is it a sample (not necessarily random) of instances from a larger set?** If the dataset is a sample, then what is the larger set? Is the sample representative of the larger set (e.g., geographic coverage)? If so, please describe how this representativeness was validated/verified. If it is not representative of the larger set, please describe why not (e.g., to cover a more diverse range of instances, because instances were withheld or unavailable).

  Answer: The dataset contains all instances from the existing benchmark datasets.

- Question: **What data does each instance consist of?** "Raw" data (e.g., unprocessed text or images) or features? In either case, please provide a description.

  Answer: Each instance consists of the transformed text, answer choices, and label.

- Question: **Is there a label or target associated with each instance?** If so, please provide a description.

  Answer: Yes, each label comes from the original QA datasets.

- Question: **Is any information missing from individual instances?** If so, please provide a description, explaining why this information is missing (e.g., because it was unavailable). This does not include intentionally removed information, but might include, e.g., redacted text.

  Answer: No, there is no information missing from individual instances.

- Question: **Are relationships between individual instances made explicit (e.g., users' movie ratings, social network links)?** If so, please describe how these relationships are made explicit.

  Answer: No.

- Question: **Are there recommended data splits (e.g., training, development/validation, testing)?** If so, please provide a description of these splits, explaining the rationale behind them.

Answer: This dataset is for testing only.

- Question: **Are there any errors, sources of noise, or redundancies in the dataset?** If so, please provide a description.

  Answer: No, we have verified that there are no errors in the datasets.

- Question: **Is the dataset self-contained, or does it link to or otherwise rely on external resources (e.g., websites, tweets, other datasets)?** If it links to or relies on external resources, a) are there guarantees that they will exist, and remain constant, over time; b) are there official archival versions of the complete dataset (i.e., including the external resources as they existed at the time the dataset was created); c) are there any restrictions (e.g., licenses, fees) associated with any of the external resources that might apply to a dataset consumer? Please provide descriptions of all external resources and any restrictions associated with them, as well as links or other access points, as appropriate.

  Answer: Our dataset is self-contained.

- Question: **Does the dataset contain data that might be considered confidential (e.g., data that is protected by legal privilege or by doctor–patient confidentiality, data that includes the content of individuals' non-public communications)?** If so, please provide a description.

  Answer: No.

- Question: **Does the dataset contain data that, if viewed directly, might be offensive, insulting, threatening, or might otherwise cause anxiety?** If so, please describe why.

  Answer: No.

### A.3   Preprocessing/cleaning/labeling

- Question: **Was any preprocessing/cleaning/labeling of the data done (e.g., discretization or bucketing, tokenization, part-of-speech tagging, SIFT feature extraction, removal of instances, processing of missing values)?** If so, please provide a description. If not, you may skip the remaining questions in this section.

  Answer:

- Question: **Was the "raw" data saved in addition to the preprocessed/cleaned/labeled data (e.g., to support unanticipated future uses)?** If so, please provide a link or other access point to the "raw" data.

  Answer: No.

- Question: **Is the software that was used to preprocess/clean/label the data available?** If so, please provide a link or other access point.

  Answer:
    - Google Sheets: https://docs.google.com/spreadsheets/
    - Python: https://www.python.org/

### A.4   Uses

- Question: **Has the dataset been used for any tasks already?** If so, please provide a description.

  Answer: No.

- Question: **Is there a repository that links to any or all papers or systems that use the dataset?** If so, please provide a link or other access point.

  Answer: No.

- Question: **What (other) tasks could the dataset be used for?**

  Answer: N/A

- Question: **Is there anything about the composition of the dataset or the way it was collected and preprocessed/cleaned/labeled that might impact future uses?** For example, is there anything that a dataset consumer might need to know to avoid uses that could result

in unfair treatment of individuals or groups (e.g., stereotyping, quality of service issues) or other risks or harms (e.g., legal risks, financial harms)? If so, please provide a description. Is there anything a dataset consumer could do to mitigate these risks or harms?

Answer: N/A

- Question: **Are there tasks for which the dataset should not be used?** If so, please provide a description.

Answer: N/A

## A.5 Distribution

- Question: **Will the dataset be distributed to third parties outside of the entity (e.g., company, institution, organization) on behalf of which the dataset was created?** If so, please provide a description.

Answer: Yes, the dataset will be made publicly accessible through Hugging Face.

- Question: **How will the dataset will be distributed (e.g., tarball on website, API, GitHub)?** Does the dataset have a digital object identifier (DOI)?

Answer: The datasets will be distributed on Hugging Face with public access.

- Question: **When will the dataset be distributed?**

Answer: The dataset is publicly available on Hugging Face since May 12, 2025.

- Question: **Will the dataset be distributed under a copyright or other intellectual property (IP) license, and/or under applicable terms of use (ToU)?** If so, please describe this license and/or ToU, and provide a link or other access point to, or otherwise reproduce, any relevant licensing terms or ToU, as well as any fees associated with these restrictions.

Answer: The datasets are distributed under the CC BY-SA 4.0 license.

- Question: **Have any third parties imposed IP-based or other restrictions on the data associated with the instances?** If so, please describe these restrictions, and provide a link or other access point to, or otherwise reproduce, any relevant licensing terms, as well as any fees associated with these restrictions.

Answer: No.

- Question: **Do any export controls or other regulatory restrictions apply to the dataset or to individual instances?** If so, please describe these restrictions, and provide a link or other access point to, or otherwise reproduce, any supporting documentation.

Answer: No.

## A.6 Maintenance

- Question: **Who will be supporting/hosting/maintaining the dataset?**

Answer: The dataset is hosted on Hugging Face.

- Question: **How can the owner/curator/manager of the dataset be contacted (e.g., email address)?**

Answer: Contact the authors of this paper via email.

- Question: **Is there an erratum?** If so, please provide a link or other access point.

Answer: No.

- Question: **Will the dataset be updated (e.g., to correct labeling errors, add new instances, delete instances)?** If so, please describe how often, by whom, and how updates will be communicated to dataset consumers (e.g., mailing list, GitHub)?

Answer: The datasets will be updated if necessary.

- Question: **If the dataset relates to people, are there applicable limits on the retention of the data associated with the instances (e.g., were the individuals in question told that their data would be retained for a fixed period of time and then deleted)?** If so, please describe these limits and explain how they will be enforced.

Answer: The dataset does not relate with people.

- Question: **Will older versions of the dataset continue to be sup-ported/hosted/maintained?** If so, please describe how. If not, please describe how its obsolescence will be communicated to dataset consumers.

  Answer: Yes.

- Question: **If others want to extend/augment/build on/contribute to the dataset, is there a mechanism for them to do so?** If so, please provide a description. Will these contributions be validated/verified? If so, please describe how. If not, why not? Is there a process for communicating/distributing these contributions to dataset consumers? If so, please provide a description.

  Answer: No, our datasets are freely available for others to use.

## B Experiment Setting

### B.1 Computer Resources

Experiments were conducted using four NVIDIA RTX A6000 GPUs and two NVIDIA A100-SXM4-80GB GPUs. Our implementation is built on vLLM (v0.5.5), PyTorch (v2.4.0), Hugging Face Transformers (v4.47.0), and Datasets (v3.1.0). On average, each dataset required approximately 10 hours for transformation.

### B.2 Computation Requirements

We used 2 NVIDIA RTX A6000 GPU to transform each dataset into targeted variants using Gemma-2-27B-Instruct [66]. Below are the time required for transformation for each dataset.

- MMLU: 11h
- ARC: 2h 50min
- TruthfulQA: 1h 50min
- GSM8K: 3h
- HellaSwag: 8h
- WinoGrande: 2h 40min

## C Dataset Construction

### C.1 English Dialects

#### C.1.1 Electronic World Atlas of Varieties of English (eWAVE)

The Electronic World Atlas of Varieties of English (eWAVE) [37] is a curated database documenting 235 linguistic features across 77 English varieties. Developed by 84 professional linguists and grounded in 175 peer-reviewed sources, eWAVE provides a structured taxonomy of features spanning 12 grammatical categories: Pronouns, Noun Phrase, Tense and Aspect, Modal Verbs, Verb Morphology, Negation, Agreement, Relativization, Complementation, Adverbial Subordination, Adverbs and Prepositions, and Discourse and Word Order. Each feature is accompanied by illustrative examples. Varieties are annotated with six levels of feature prevalence: (i) feature is pervasive or obligatory, (ii) feature is neither pervasive nor extremely rare, (iii) feature exists, but is extremely rare, (iv) attested absence of feature, (v) feature is not applicable (given the structural make-up of the variety/P/C), and (vi) no information on feature is available.

#### C.1.2 Dialect Selection

We first mapped the presence strength of each feature per dialect to one of four discrete levels.

- feature is pervasive or obligatory: 1.0
- feature is neither pervasive nor extremely rare: 0.5
- feature exists, but is extremely rare: 0.25

- attested absence of feature, feature is not applicable, no information on feature is available: 0

We then applied Singular Value Decomposition (SVD) for dimensionality reduction, retaining 90% of the variance. Using the reduced feature representations for each dialect, we performed K-Nearest Neighbors (KNN) clustering with the number of clusters set to 5. The choice of 5 clusters was informed by both the Elbow Method and Silhouette Scores, which indicated that 5 was the most optimal number of clusters. Then we selected clusters with famous English dialects such as African American Vernacular English and Welsh English. The final 18 dialects and their abbreviations are as follows: African American Vernacular English (AAVE), Irish English (IrE), Australian English (AuE), Bahamian English (BahE), East Anglian English (EAngE), Appalachian English (AppE), English dialects in the Southeast of England (SE-Eng), Australian Vernacular English (AuE-V), English dialects in the North of England (NE-Eng), English dialects in the Southwest of England (SW-Eng), Manx English (Manx), New Zealand English (NZE), Newfoundland English (NfE), Ozark English (OzE), Scottish English (ScE), Southeast American enclave dialects (SE-AmE), Tristan da Cunha English (TdCE), Welsh English (WeE).

## C.2 ESL English-L1

### C.2.1 Number of Samples in Compiled Dataset.

Table 7 shows the number of samples per L1 and per CEFR level collected from three learner corpora: CLC-FCE [72], ICLE [25], and EFCamDat [21].

Table 7: Number of samples collected from CLC-FCE, ICLE, and EFCamDat.

| | CLC-FCE | | ICLE | | EFCamDat | | Total | |
|---|---|---|---|---|---|---|---|---|
| | A | B | A | B | A | B | A | B |
| Arabic | 0 | 0 | 0 | 0 | 24,155 | 4,857 | 24,155 | 4,857 |
| Chinese-Mandarin | 9 | 107 | 1 | 45 | 106,654 | 22,289 | 106,664 | 22,441 |
| French | 2 | 245 | 0 | 0 | 22,244 | 9,646 | 22,246 | 9,891 |
| German | 2 | 120 | 3 | 42 | 25,040 | 14,501 | 25,045 | 14,663 |
| Italian | 2 | 121 | 1 | 8 | 22,787 | 11,672 | 22,790 | 11,801 |
| Japanese | 6 | 134 | 10 | 171 | 11,653 | 5,081 | 11,669 | 5,386 |
| Portuguese | 1 | 114 | 1 | 43 | 248,200 | 61,751 | 248,202 | 61,908 |
| Russian | 10 | 134 | 0 | 12 | 35,081 | 13,287 | 35,091 | 13,433 |
| Spanish | 16 | 351 | 6 | 47 | 52,786 | 11,456 | 52,808 | 11,854 |
| Turkish | 8 | 126 | 0 | 61 | 7,899 | 2,237 | 7,907 | 2,424 |

### C.2.2 CEFR Pseudo-Label Generation

The CLC-FCE and ICLE datasets do not include annotated CEFR levels. To address this, we employed `gpt-4o-mini-2024-07-18` to generate pseudo-CEFR labels. The prompt used for label generation is provided in Table 8.

Table 8: Prompt used for pseudo CEFR label generation.

**System:**
You are a linguistic expert.

**User:**
Classify the given sentence among three CEFR levels (A, B, C). Respond only CEFR level.
Sentence: {sentence}

### C.2.3 Outputs from the Automatic Grammar Checker

The outputs from the automatic grammar checker are overly specific, identifying narrow error types such as "I told her (to) break a leg" or "this render (renders) the . . . ". To enable more effective

analysis, we consolidated similar low-level errors into broader categories. For instance, the category "Omission of a Preposition" includes examples like "I told her (to) break a leg" and "It would be great (to) write a story." The category "Mismatch between Article and Noun" captures cases such as "I like to use a pens and paper," "I have received a 150 likes," and "The cat is an animals."

In total, we define 42 higher-level categories: "Confusion between effects and affects", "Double negation", "Gerund complement after psych/perception verb", "Inappropriate formulaic closing", "Incorrect existential agreement with plural noun", "Incorrect passive voice usage", "Incorrect pluralization after 'either of' ", "Incorrect use of 'if' instead of 'whether' ", "Incorrect use of gerund after 'advise' ", "Incorrect verb usage with auxiliary", "Mismatch between article and noun", "Mismatch between noun and adjective", "Mismatch between subject and verb", "Missing complementizer 'to' after 'allow' ", "Missing determiner after quantifier", "Misusage of irregular past tense verbs", "Misuse of 'have' and 'having' ", "Non-standard negation with 'let's' ", "Omission of a preposition", "Omission of a verb", "Omission of object pronoun", "Omission of required articles", "Omission of subject", "Plural noun required after quantifier phrase", "Redundant discourse marker usage", "Redundant modal construction", "Redundant phrase repetition", "Redundant verb in question form", "Singular form in fixed polite expression", "Usage of 'couple times' instead of 'a couple of times' ", "Usage of a plural noun when a singular form is required", "Usage of a plural noun where a singular is required after 'is there any' ", "Usage of a singular noun when a plural form is required", "Usage of an adjective where an adverb is required", "Usage of an auxiliary verb when unnecessary", "Usage of an incorrect past participle form", "Usage of first-person subject with 'according to' ", "Usage of passive voice when active voice is required' ", "Usage of plural auxiliary 'do' with singular subject 'anyone' ", "Use of 'much' with countable noun", "Use of continuous aspect with stative verbs", "Use of plural noun with each/every."

### C.2.4  L1-Specific Features

The following are the extracted features categorized by L1.

- Arabic: Usage of a plural noun where a singular is required after 'is there any', Incorrect passive voice usage, Usage of 'couple times' instead of 'a couple of times', Omission of a preposition, Mismatch between article and noun, Omission of a verb, Usage of a singular noun when a plural form is required, Omission of subject, Missing determiner after quantifier, Mismatch between article and noun

- Chinese-Mandarin: Usage of plural auxiliary 'do' with singular subject 'anyone', Inappropriate formulaic closing, Mismatch between subject and verb, Singular form in fixed polite expression, Omission of subject, Usage of an incorrect past participle form, Mismatch between article and noun, Incorrect existential agreement with plural noun, Usage of passive voice when active voice is required

- French: Non-standard negation with 'let's', Usage of 'couple times' instead of 'a couple of times', Redundant verb in question form, Misuse of 'have' and 'having', Usage of a plural noun where a singular is required after 'is there any', Use of plural noun with each/every, Gerund complement after psych/perception verb, Omission of a preposition, Omission of a verb, Usage of first-person subject with 'according to'

- German: Incorrect passive voice usage, Usage of 'couple times' instead of 'a couple of times', Misuse of 'have' and 'having', Gerund complement after psych/perception verb, Omission of a preposition, Incorrect verb usage with auxiliary, Misusage of irregular past tense verbs, Use of 'much' with countable noun, Usage of an adjective where an adverb is required, Incorrect use of gerund after 'advise'

- Italian: Incorrect use of 'if' instead of 'whether', Usage of 'couple times' instead of 'a couple of times', Usage of a plural noun where a singular is required after 'is there any', Redundant discourse marker usage, Incorrect pluralization after 'either of', Gerund complement after psych/perception verb, Use of plural noun with each/every, Usage of a singular noun when a plural form is required, Omission of a verb, Misusage between 'not' and 'never'

- Japanese: Use of continuous aspect with stative verbs, Mismatch between noun and adjective, Redundant modal construction, Usage of a singular noun when a plural form is required, Omission of a preposition, Gerund complement after psych/perception verb, Missing determiner after quantifier, Plural noun required after quantifier phrase, Omission of required articles, Omission of object pronoun

- Portuguese: Omission of a preposition, Omission of subject, Gerund complement after psych/perception verb, Usage of an auxiliary verb when unnecessary, Usage of a singular noun when a plural form is required, Missing complementizer 'to' after 'allow', Singular form in fixed polite expression, Redundant phrase repetition, Double negation, Incorrect existential agreement with plural noun

- Russian: Redundant verb in question form, Mismatch between article and noun, Misusage of preposition, Mismatch between subject and verb, Omission of a verb, Omission of subject, Missing complementizer 'to' after 'allow', Omission of a preposition, Redundant verb, Redundant preposition

- Spanish: Non-standard negation with 'let's', Incorrect pluralization after 'either of', Mismatch between article and noun, Omission of subject, Omission of a preposition, Incorrect verb usage with auxiliary, Usage of a singular noun when a plural form is required, Missing Determiner after Quantifier, Redundant verb, Misusage of article in uncountable noun

- Turkish: Confusion between effects and affects, Usage of first-person subject with 'according to', Usage of a singular noun when a plural form is required, Omission of a preposition, Missing complementizer 'to' after 'allow', Omission of subject, Usage of a plural noun when a singular form is required, Missing determiner after quantifier, Mismatch between article and noun, Redundant adverb

### C.3 Transformation Guideline Generation

We use `gpt-4-0613` to generate transformation guidelines via one-shot prompting, with a temperature of 0.8 and top-$p$ sampling set to 0.95. The model is provided with the name of the linguistic feature, a brief description, and representative examples. It is then instructed to (1) describe the linguistic characteristics of the feature, and (2) outline a step-by-step transformation procedure consisting of two phases: *Qualification*, which checks whether the feature applies to a given sentence, and *Application*, which modifies the sentence accordingly.

We emphasize that the transformation process should focus strictly on lexical rules, avoiding subjective elements such as emotional or cultural interpretation, metaphor, or judgments of significance. The full prompt used for generating transformation guidelines is shown in Table 9, and examples of the resulting guidelines are presented in Table 10.

### C.4 Transforming into English Varieties

### C.4.1 Transformation of Vocabulary into Target CEFR Levels

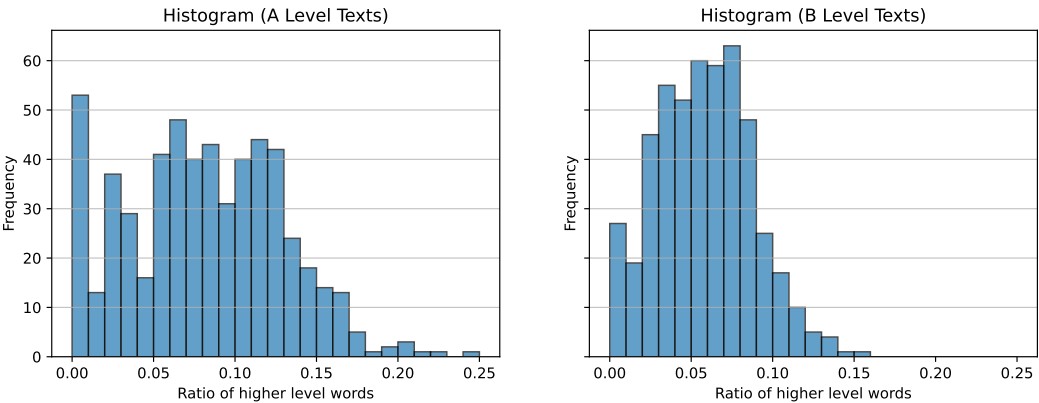

Figure 3: Histograms of distributions of higher-level word usage in CEFR A and CEFR B texts.

To ensure that the transformed outputs for ESL English varieties reflect realistic proficiency levels, we incorporated a vocabulary substitution step guided by CEFR-level annotations. To acknowledge that ESL learners often know a small fraction of advanced words even at lower proficiency levels, we first analyze a CEFR-labeled English text dataset to find out the ratio of higher-level words used by

Table 9: Prompt for generating transformation guidelines

**System:**
You are a linguistic expert. I am a student trying to understand the given linguistic feature and transform a sentence reflecting the feature. As an expert, first explain the linguistic characteristics of the given linguistic feature. Then, outline detailed steps to transform a given sentence to reflect the characteristic, breaking the process into two main phases: Qualification and Application. Qualification refers to steps that identify if the linguistic feature is applicable to the given sentence in yes/no question format where answers to all questions should be 'yes' if feature dialect is applicable. Here, questions in qualification should be strictly related to lexical rules and should not ask for any decisions asking for significance or emotions. The questions should also not include questions about context, culture, or metaphors. Application refers to the action items that a model should take in order to reflect the given linguistic feature. All questions and action items should strictly be related to lexicon. All questions and action items should not include context, culture, or metaphor where answers might differ by people such as determining significant role or emotional attachment.

**User:**
Linguistic Feature: She/her used for inanimate referents.
Description: Pronouns, pronoun exchange, nominal gender.
Examples: The boat I had, was a seventy-two foot boat., She was built in Joneses-Slip in nineteen-fifty-five by my father., She's a nice bike., I left the boat there as she was., This is another pot and saucer., A bit dusty! You see that one isn't exactly glazed proper, burnt proper is she?, She's a twenty point five metre sloop again designed by Kel Steinman an Australian, So the Bransfield is the BAS ship then? Yeah she was the sort of kingpin until they brought the James Clark Ross

**Assistant:**
Linguistic Characteristic: She/Her Used for Inanimate Referents
The phenomenon of using 'she' or 'her' for inanimate objects is primarily influenced by cultural associations, personification, or grammatical gender in certain languages. In English, it is less common but can occur in specific contexts, such as referring to ships, cars, or countries, often reflecting affection, personification, or historical conventions. This use can evoke an emotional connection or imply a particular viewpoint about the inanimate referent.

Steps to Transform a Sentence: To transform a given sentence to reflect the use of 'she/her' for inanimate referents, we can break the process down into two main phases: Qualification and Application.
Qualification
1. Is there an inanimate referent in the sentence?
2. Does the original sentence contain a pronoun that can be replaced with 'she' or 'her'?
If the answers to all relevant questions are "Yes," then the linguistic feature is applicable.
Application
1. Identify the inanimate referent in the sentence that will be modified.
2. Replace any pronouns referring to the inanimate referent with "she" or "her."

**User:**
Linguistic Feature: {linguistic feature}
Description: {description of linguistic feature}
Examples: {examples of linguistic feature}

---

lower CEFR proficiency level learners, as mentioned in Section 3.3. Figure 3 presents the distribution of higher-level vocabulary in the dataset (*e.g.*, , B or C level words in A level texts). Notably, for both target levels A and B, at least 90% of the samples contain no more than 15% of vocabulary from higher CEFR levels than the designated target level. This empirical finding motivated our decision to allow up to 15% of higher-level vocabulary in transformed outputs. This threshold balances fidelity to learner-level constraints with linguistic realism, acknowledging that ESL learners often know a small fraction of advanced words even at lower proficiency levels.

Table 11 shows the prompt used for finding pseudo-labels for words without a CEFR label in the Oxford vocabulary lists, and Table 12 presents the prompt used for transforming higher-level vocabulary in a sentence to a target level. The value of `min_transform_words` is set to 15% of the total word count in `question_text` and serves as the threshold for permitted higher-level words.

Table 10: Examples of transformation guidelines.

**Feature: Myself/meself instead of I in coordinate subjects**

**Qualification**:
1. Is there a coordinate subject in the sentence? A coordinate subject is formed when two subjects are joined by a conjunction like 'and' or 'or'.
2. Does the coordinate subject include 'I'?

If the answers to all relevant questions are 'Yes', then the linguistic feature is applicable.

**Application**:
1. Identify the coordinate subject in the sentence that includes 'I'.
2. Replace 'I' with 'myself' in the coordinate subject.

**Feature: Omission of Required Articles**

**Qualification:**
1. Does the sentence contain a noun that requires an article ('a', 'an', or 'the') for grammatical correctness or clarity?
2. Is the noun countable and in singular form, or does it refer to something specific that needs 'the'?

If the answers to all relevant questions are 'Yes', then the linguistic feature is applicable.

**Application:**
1. Identify the noun(s) that require an article for grammatical correctness.
2. Remove the article ('a', 'an', or 'the') preceding the noun or leave the noun without any article.

Table 11: Vocabulary pseudo-label prompt.

**System:** You are an expert in classifying vocabulary into CEFR levels. Given a single word, classify it into its appropriate CEFR level when used with its most common definition. If it is a proper noun, answer with A1. Answer only with one of the following: A1, A2, B1, B2, C1, C2.

**User:** {word}

Table 13 presents the transformation success rates by CEFR level and dataset, showing how often our pipeline was able to produce outputs that met CEFR-level vocabulary constraints while preserving semantic equivalence.

### C.4.2 Vocabulary Threshold Experiment

We conducted experiments varying vocabulary threshold using TruthfulQA [42] with LLaMA-3.3-70B-Instruct [26]. The result is in Table 14. The 15% threshold was used in the main paper. As shown in the table, a stricter threshold (5%) leads to a decline in performance, whereas a more lenient threshold (25%) results in improved performance across all variants. We hypothesize that a stricter threshold compels the model to replace a greater number of vocabulary items with simpler alternatives, thereby increasing the degree of transformation from the original sentence and potentially compromising meaning or coherence.

### C.5 Prompts used for Transformation

Table 15 presents the one-shot prompt used to transform a Standard American English (SAE) sentence $s$ into a target variety using the feature transformation model $T$. Each transformation is guided by a feature-specific guideline and example. The model is instructed to follow the guideline strictly, preserving the structure and core meaning of the original sentence while disregarding grammatical correctness.

To ensure semantic fidelity, we employ a semantic checker model $S$ using a zero-shot prompt, as shown in Table 16. The verification process emphasizes the preservation of key content elements such as keywords, numerical information, and core propositions, while ignoring minor grammatical deviations, including incorrect or missing prepositions and redundancy.

Table 12: Vocabulary transformation prompt.

**System:** You are an expert in transforming vocabulary of higher CEFR levels to level {target_level}. You are given higher level words that appear in the question: {words_to_transform}. Please replace at least {min_transform_words} words with synonyms in level {target_level}.

**User:** {question_text}

Table 13: Number and ratio of valid vocabulary transformations by dataset.

| Dataset | Size | Target CEFR | Valid Transf. | Transf. Ratio |
|---------|------|-------------|---------------|---------------|
| MMLU | 14042 | A | 7246 | 51.6% |
| | | B | 11970 | 85.2% |
| GSM8K | 1319 | A | 1219 | 92.4% |
| | | B | 1315 | 99.7% |
| ARC | 1172 | A | 774 | 66.0% |
| | | B | 1132 | 96.6% |
| HellaSwag | 10042 | A | 7593 | 75.6% |
| | | B | 9903 | 98.6% |
| TruthfulQA | 817 | A | 623 | 76.3% |
| | | B | 781 | 95.6% |
| WinoGrande | 1267 | A | 945 | 74.6% |
| | | B | 1247 | 98.4% |

## C.6 Transformation Ratio

Tables 17 and 18 report the average number of features applied per sample and the overall proportion of transformed samples for dialect and ESL English, respectively, as discussed in Section 3.4. Consistent with the results presented in the main paper, ESL English exhibits a higher transformation rate and a greater average number of features applied per sample compared to dialects.

## C.7 Human Evaluation

Human annotators were shown one sample at a time, with a total of 150 samples randomly shuffled, 25 from each model. For each sample, annotators answered two binary (yes/no) questions: (Q1) whether the model correctly followed the Qualification and Application steps specified in the transformation guideline, and (Q2) whether the transformed sentence preserved the original meaning. The interface presented to annotators is shown in Figure 4. A sample was considered valid if it received majority approval from the annotators.

# D Experiments

## D.1 Experiment Setting

We evaluated the transformed datasets on seven state-of-the-art models: Qwen2.5-72B-Instruct [67], DeepSeek-R1-Distill-Llama-70B [14], LLaMA-3.3-70B-Instruct [26], Gemini 2.0 Flash [23], Gemini 2.5 Pro [22], GPT-4o-mini [50], and o4-mini [52].[16] We set the maximum number of generated tokens to 2048 and conducted all experiments in a zero-shot setting. The system prompt used was: "Do not reason for too long. If the question is a multiple choice question, answer with the option letter. If none of the given options match, you may guess or say 'none of the above.' Start your final sentence with 'The answer is '." To extract the model's prediction, we parsed the output beginning from the phrase "The answer is", using the subsequent text as the final answer.

---

[16]Model versions: gemini-2.5-pro-exp-03-25, gpt-4o-mini-2024-07-18, o4-mini-2025-04-16

Table 14: Experiment results varying vocabulary threshold.

|  | A_arabic | A_italian | B_arabic | B_italian |
|---|---|---|---|---|
| 15% (Reported in the paper) | 63.2 | 65.3 | 63 | 66.8 |
| 25% | 63.8 | 66.8 | 65.3 | 67.4 |
| 5% | 60.4 | 63.2 | 62.1 | 64.2 |

Table 15: Prompt for transforming into varieties.

**System:** Your task is to rephrase the given sentence by following the guideline. {transformation guideline}

1. **Qualification**:
- Answer the qualification questions for the linguistic feature with either "yes" or "no."
- Answer the questions in a very strict manner.
- Proceed to the next step only if **all** answers are "yes."
- Otherwise, stop in qualification phase with generating '**Transformed Sentence:** (No change)'.

2. **Application**:
- Make only the **necessary changes** to apply the linguistic feature, ensuring no loss of information.
- Provide the final transformed sentence, adhering strictly to the format and structure of the given example.

### Mandatory
- Proceed to Application only if all answers to the qualification questions are 'yes'.
- Preserve the structure of the original sentence as much as possible with no information loss.
- Follow the guideline, not considering standard English grammar.
- Final sentence should start with '**Transformed Sentence:**' either with sentence of (No change).

**User:** **Original Sentence**: {example sentence}

**Assistant:** {example output}

**User:** **Original Sentence**: {SAE written sentence}

## D.2 Full Experiment Analysis

Figures 5 and 6 present the full analysis results across all datasets, corresponding to the analysis in Section 4.3. Consistent with the findings in the main paper, we observe a positive correlation between linguistic distance from Standard American English (SAE) and performance degradation, although the strength of this relationship varies across datasets. In ESL English, despite some deviations, performance drop generally increases with the difficulty level of the English variety.

## D.3 Full Correlation Values

We conducted three statistical tests: pearson correlation coefficient [5], spearman rank correlation [73], kendall's tau [9], and the results are in Table 19. The results show that all p-values, except for HellaSwag under the pearson correlation coefficient, are below 0.05, indicating statistically significant linear correlations between performance gaps and linguistic distances.

For ESL variants, we performed a one-way ANOVA [57] to examine whether there are statistically significant differences among the four L1 groups. The results are in Table 20. Across all datasets, the p-values are extremely low, indicating clear performance differences among the four L1 groups.

Table 16: Prompt for semantic check.

**User:** Determine whether the meaning of Sentence 1 is significantly altered or lost in Sentence 2.

### Consideration
- All keywords from Sentence 1 should be in Sentence 2.
- All numbers in Sentence 1 should match with Sentence 2.
- Focus on core information only.
- Ignore grammar; it is not a factor for consideration.
- Missing or incorrect prepositions should not be considered.
- Ignore repetition of phrases. Repetition is not a factor for consideration.
- Base your decision solely on whether essential information is missing.

Respond with either 'yes' or 'no' only.

Sentence 1: {SAE written sentence}
Sentence 2: {transformed sentence}

Answer:

Table 17: Average number of features applied per sample and proportion of transformed samples in dialect.

|  | MMLU | ARC | TruthfulQA | GSM8K | Hellaswag | WinoGrande |
|---|---|---|---|---|---|---|
| AAVE | 1.12 / 61.8% | 1.17 / 65.1% | 0.76 / 45.0% | 0.80 / 54.7% | 2.01 / 87.4% | 2.06 / 88.4% |
| AppE | 1.53 / 70.6% | 1.14 / 64.9% | 1.08 / 60.5% | 1.11 / 63.4% | 2.26 / 88.9% | 2.70 / 96.5% |
| AuE | 0.80 / 65.3% | 0.76 / 64.8% | 0.49 / 41.5% | 0.40 / 33.0% | 0.91 / 66.5% | 1.60 / 96.8% |
| AusVE | 0.95 / 57.5% | 0.76 / 50.7% | 0.78 / 57.8% | 1.05 / 70.5% | 1.53 / 82.9% | 1.63 / 91.9% |
| BahE | 2.63 / 70.5% | 1.94 / 53.7% | 1.76 / 63.4% | 2.91 / 76.6% | 3.20 / 83.9% | 6.22 / 99.5% |
| EAngE | 3.54 / 87.7% | 3.08 / 86.1% | 2.87 / 90.0% | 3.75 / 90.2% | 4.58 / 95.9% | 5.94 / 99.8% |
| IrE | 2.67 / 91.0% | 2.92 / 95.0% | 2.49 / 87.8% | 1.80 / 78.8% | 4.82 / 98.9% | 4.53 / 100.0% |
| Manx | 1.86 / 86.8% | 1.64 / 86.9% | 1.57 / 80.7% | 0.84 / 60.5% | 2.57 / 95.8% | 3.22 / 98.3% |
| NE-Eng | 0.70 / 59.6% | 0.77 / 70.5% | 0.43 / 38.8% | 0.58 / 54.7% | 1.43 / 89.9% | 1.05 / 77.0% |
| NZE | 2.07 / 84.7% | 2.12 / 88.2% | 1.48 / 70.3% | 2.15 / 85.8% | 3.10 / 97.3% | 3.48 / 99.4% |
| NfE | 4.17 / 95.4% | 3.98 / 96.4% | 3.31 / 92.5% | 4.3 / 96.7% | 5.55 / 98.9% | 7.63 / 99.9% |
| OzE | 2.50 / 86.6% | 2.73 / 91.9% | 2.17 / 85.8% | 2.75 / 89.8% | 3.59 / 96.6% | 4.07 / 99.2% |
| SE-AmE | 2.50 / 79.9% | 2.19 / 70.8% | 2.03 / 79.1% | 2.98 / 84.9% | 3.65 / 91.4% | 4.72 / 99.6% |
| SE-Eng | 0.20 / 17.4% | 0.14 / 13.2% | 0.07 / 6.6% | 0.22 / 19.7% | 0.26 / 22.9% | 0.30 / 25.7% |
| SW-Eng | 0.90 / 66.3% | 0.77 / 62.9% | 0.55 / 43.6% | 0.33 / 30.0% | 0.84 / 64.4% | 1.67 / 96.9% |
| ScE | 1.15 / 69.8% | 1.06 / 67.9% | 1.06 / 63.5% | 0.76 / 51.2% | 1.20 / 70.3% | 2.05 / 97.8% |
| TdCE | 0.94 / 44.9% | 0.69 / 35.9% | 0.47 / 31.1% | 1.12 / 54.7% | 2.17 / 85.8% | 2.10 / 92.9% |
| WeE | 2.27 / 90.1% | 2.11 / 89.9% | 2.51 / 97.1% | 1.08 / 61.0% | 1.81 / 83.5% | 3.01 / 98.7% |

Table 18: Average number of features applied per sample and proportion of transformed samples in ESL English.

|   |    | MMLU | ARC | TruthfulQA | GSM8K | Hellaswag | WinoGrande |
|---|----|------|-----|------------|-------|-----------|------------|
| A | ar | 2.65 / 96.6% | 2.77 / 98.8% | 2.05 / 92.6% | 3.06 / 99.5% | 2.88 / 98.2% | 2.87 / 99.8% |
|   | de | 2.17 / 93.4% | 2.30 / 94.8% | 1.92 / 91.8% | 2.15 / 94.9% | 2.88 / 96.0% | 2.98 / 99.7% |
|   | es | 3.15 / 97.1% | 3.50 / 99.7% | 2.74 / 97.1% | 3.53 / 99.3% | 3.55 / 98.3% | 3.63 / 99.6% |
|   | fr | 1.00 / 84.6% | 0.99 / 86.6% | 0.83 / 74.8% | 1.15 / 87.2% | 1.16 / 87.6% | 1.11 / 92.8% |
|   | it | 1.03 / 80.8% | 1.05 / 87.5% | 0.75 / 68.1% | 1.20 / 87.3% | 1.33 / 89.6% | 1.19 / 87.7% |
|   | ja | 3.21 / 96.5% | 3.41 / 98.8% | 2.54 / 94.2% | 3.20 / 98.1% | 3.93 / 98.1% | 3.83 / 100.0% |
|   | pt | 2.92 / 98.1% | 3.07 / 99.5% | 2.89 / 99.4% | 3.30 / 99.8% | 3.27 / 98.3% | 3.36 / 99.9% |
|   | ru | 3.02 / 97.5% | 3.28 / 99.7% | 2.85 / 99.0% | 3.53 / 99.5% | 3.33 / 98.6% | 3.56 / 99.9% |
|   | tr | 2.94 / 96.9% | 3.08 / 97.9% | 2.34 / 92.9% | 3.29 / 98.0% | 3.18 / 97.6% | 3.22 / 99.9% |
|   | zh | 1.63 / 88.0% | 1.67 / 90.6% | 1.23 / 83.6% | 2.02 / 93.6% | 1.77 / 89.6% | 1.84 / 93.2% |
| B | ar | 2.83 / 96.4% | 2.82 / 98.2% | 2.09 / 91.5% | 3.15 / 98.5% | 2.84 / 98.8% | 2.89 / 99.2% |
|   | de | 2.09 / 92.3% | 2.01 / 91.9% | 1.95 / 91.8% | 1.96 / 91.5% | 2.54 / 94.7% | 2.98 / 99.7% |
|   | es | 3.27 / 97.4% | 3.43 / 98.9% | 2.89 / 97.4% | 3.59 / 98.9% | 3.30 / 98.8% | 3.51 / 99.9% |
|   | fr | 0.97 / 83.0% | 0.91 / 82.3% | 0.79 / 70.2% | 1.05 / 84.2% | 1.01 / 77.8% | 1.11 / 89.2% |
|   | it | 0.93 / 73.4% | 0.87 / 73.6% | 0.65 / 56.2% | 1.07 / 79.4% | 1.30 / 87.8% | 1.21 / 88.5% |
|   | ja | 3.31 / 96.6% | 3.16 / 98.0% | 2.53 / 92.4% | 3.06 / 97.9% | 3.52 / 98.3% | 3.82 / 99.8% |
|   | pt | 2.95 / 98.2% | 3.05 / 99.0% | 2.91 / 99.0% | 3.25 / 99.5% | 2.94 / 97.2% | 3.29 / 99.8% |
|   | ru | 3.15 / 97.5% | 3.33 / 99.6% | 3.01 / 98.6% | 3.59 / 99.3% | 3.02 / 97.7% | 3.52 / 99.9% |
|   | tr | 3.06 / 96.2% | 2.96 / 96.8% | 2.24 / 88.2% | 3.26 / 98.5% | 3.00 / 98.3% | 3.19 / 99.9% |
|   | zh | 1.83 / 93.3% | 1.94 / 97.3% | 1.44 / 88.3% | 2.2 / 97.8% | 1.97 / 96.1% | 2.04 / 96.6% |

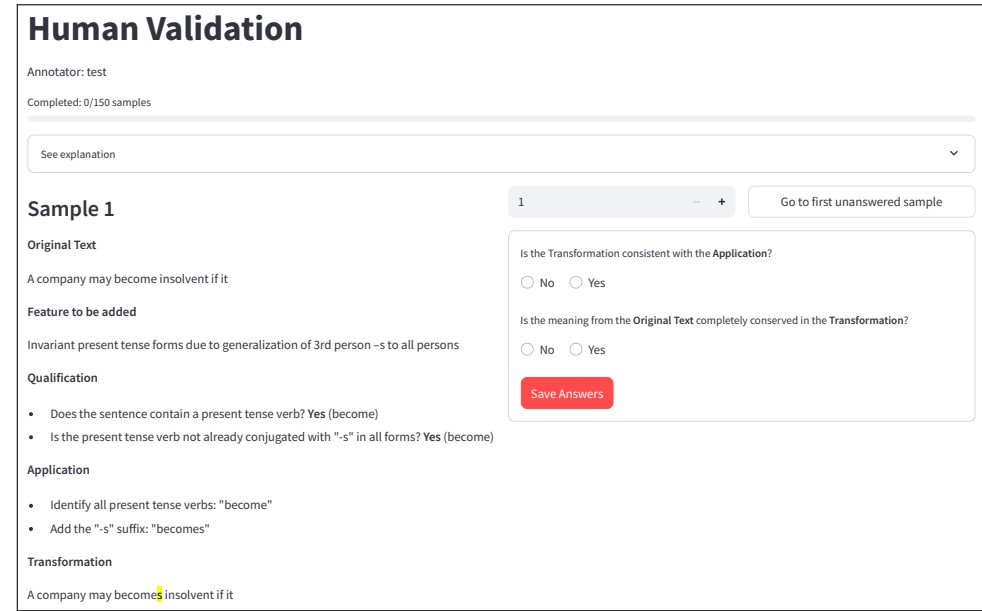

Figure 4: Interface used for human evaluation.

Table 19: Correlation values for English dialects.

| Datset | Pearson Correlation coefficient | | Spearman Rank Correlation | | Kendall's Tau | |
|--------|-------------|---------|-------------|---------|-------------|---------|
|        | coefficient | p-value | coefficient | p-value | coefficient | p-value |
| MMLU | 0.215 | 0.025 | 0.193 | 0.044 | 0.133 | 0.045 |
| ARC | 0.328 | 0.0001 | 0.354 | 4.67e-05 | 0.241 | 9.79e-05 |
| TruthfulQA | 0.292 | 0.0008 | 0.286 | 0.001 | 0.198 | 0.001 |
| GSM8k | 0.321 | 0.0002 | 0.226 | 0.010 | 0.149 | 0.015 |
| Hellaswag | 0.118 | 0.222 | 0.207 | 0.030 | 0.141 | 0.033 |
| WinoGrande | 0.429 | 5.05e-07 | 0.438 | 2.78e-07 | 0.271 | 1.04e-05 |

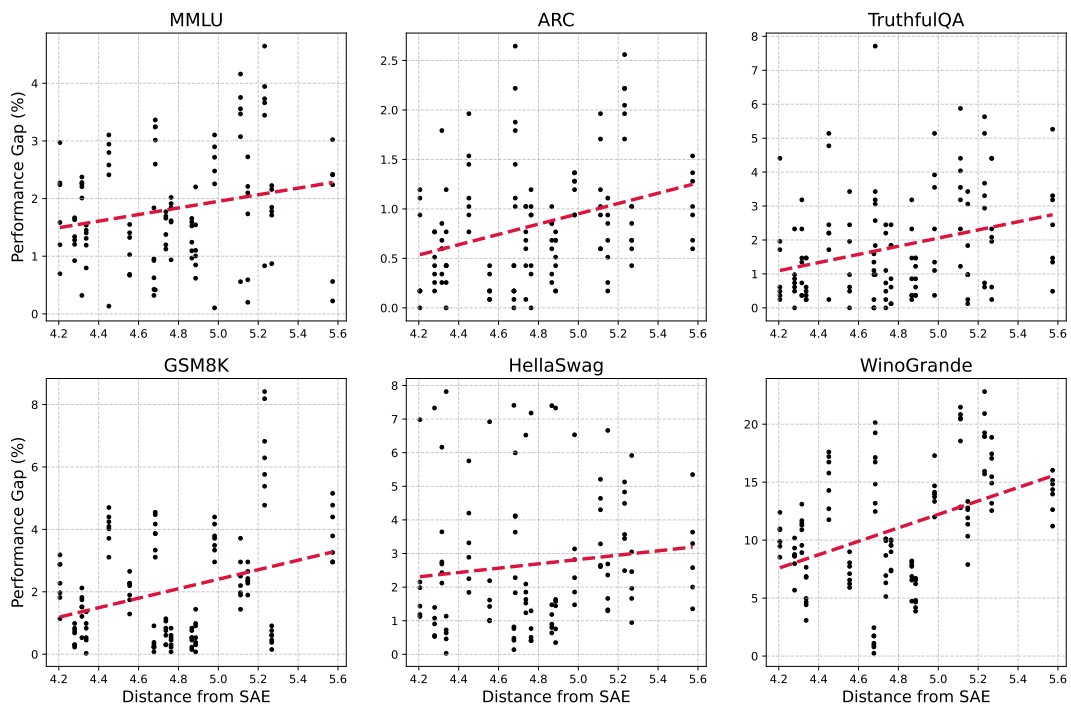

Figure 5: Correlation between linguistic distance and model performance degradation.

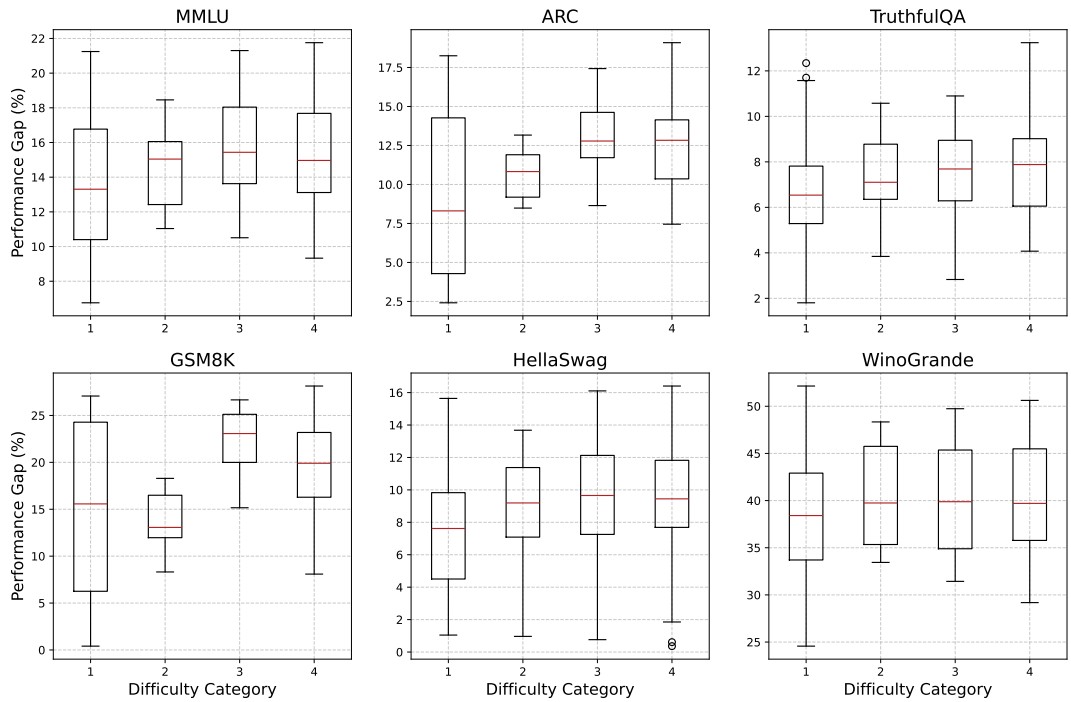

Figure 6: Boxplot by difficulty category and model performance degradation.

Table 20: Correlation values for ESL English.

| Dataset | f-stat | p-value |
|---|---|---|
| MMLU | 36.06 | 9.40e-10 |
| ARC | 92.43 | 1.23e-14 |
| TruthfulQA | 25.91 | 3.12e-08 |
| GSM8K | 160.28 | 9.84e-18 |
| Hellaswag | 62.16 | 1.69e-12 |
| WinoGrande | 43.67 | 1.08e-10 |

