# OpenReview forum: "Trans-EnV: A Framework for Evaluating the Linguistic Robustness of LLMs Against English Varieties"
_NeurIPS.cc/2025/Datasets_and_Benchmarks_Track — NeurIPS 2025 Datasets and Benchmarks Track poster_

### Official Review · Reviewer_E6DB · 2025-06-01

**Rating:** 4
**Confidence:** 4

**Summary:**

This paper proposes the first framework that converts benchmark datasets written in standard English into diverse non-standard English variants in terms of dialects and ESL English varieties by leveraging LLM sentence transformers, guided by expert-confirmed transformation guidelines. The authors observed performance gaps between standard English and non-standard English variants, especially in ESL English varieties.

**Additional Feedback:**

(minor comments on presentation)
* **Figure 1: caption of Figure 1 lacks self-completeness and can be improved further.**
  * For Fig1-(a) please consider mentioning that LLM generates transformation guidelines, and for Fig1-(b), consider explaining that capital S and T are arbitrary LLMs that take [XX] as input to produce output [YY]
  * The final transformed sentence is also notated as $s$  -- the same notation with pre-tranformed sentence $s$, which decreases the rigor.
* **Table 4 and Table 5: confusing cell colorization.**
  * I think the most important thing to emphasize here is "how the linguistic variation drops the performance of LLM compared to that of the original standard English?". It would be better to pick the performance on the original dataset as a pivot of the colormap (white), and denote the relative improvement and degeneration as blue and orange, respectively. And you can drop the bold font for simplicity.
  * The current color gradation confuses the readers about the relative performance of the converted datasets (in a quick view, blue cells seem to be improvements), which weakens the inspiring findings of Tables 4 and 5.

---

### I am open to discussion with authors. Please refute my review if there are any misunderstandings.

**Dataset Code Accessibility:**

Yes

**Ethical Considerations:**

No, there are no or only very minor ethics concerns

**Final Justification:**

Through the rebuttal, the authors provided sufficient empirical evidence that addresses my initial concerns on "limited scope of validation" and "lack of rigor in statistical test". I believe concerns from other reviewers were also professionally addressed through extensive experiments. Although this paper in its submitted form does not contain rigorous details behind the human study, I believe the strengths of this work outweigh my unresolved concern which is why I changed my rating from 3 to 4, leaning towards acceptance.

**Limitations Weaknesses:**

* [`W1`] **Limited scope of validation**
  - `W1-1`) Although the author provided demonstrations on six NLP datasets, all of them are traditional closed-form tasks that have a ground-truth discrete answer per query.
    - However, research communities may be (much more) interested in the results on open-ended setups with human preference evaluation, such as [1,2,3], rather than closed-ended answers to reflect complex real-world use cases and be suitable for agentic-LLM evaluation.
  - `W1-2`) Besides, given that the transformations are conducted at the sentence level, the length of sentences would be an important factor that affects the distributional discrepancy between the original and converted. Adding an analysis on the performance gap at varying average sentence lengths would deepen the insights from this benchmarking setup.
* [`W2`] **Lack of rigor in statistical tests and their significance**
  - `W2-1`) In Figure 2, the author claims correlations between the performance gaps and linguistic distances between the two datasets. However, the author did not provide any actual value, e.g., correlation coefficients from some statistical tests and corresponding p-values, and claimed the correlation based on just scatter plot (or barplot) visualizations, where the tendencies are unclear enough.
    - Explicit correlation analysis between performance gap and linguistic distance [4] should be accompanied to justify the statement.
  - `W2-2`) Meanwhile, in line 153-154, the author said they identified distinct features that are attributed to varying native language-proficiency levels via t-test.
    - However, the t-test is a statistical test that scrutinizes the mean difference between two groups [5], rather than identifying explanatory variables that affect the variance of the response variable. That is, t-test does not tell the individual features' importance to the variance of the response variable (language proficiency). To my knowledge, Logistic Regression (and seeing fitted coefficients) or ANCOVA would be valid statistical methods for this.
    - I would appreciate it if the authors could clarify for which variables and for what purpose the t-test was applied here.
* [`W3`] **Lack of rigor in human study**
  - `W3-1`) Claiming validity based on domain experts' oversight without detailed rationales.
    - In line 120-121, line 156-157, and line 174, the authors said like "was verified by the specialist, ..., were reviewed by the researcher" without any justification on (1) _how did that expert make the verifications?_ and (2) _based on what criteria?_.
    - Missing justification on the above important facets weakens the rigor of the proposal.
  - `W3-2`) Missing details on the recruiting process of the participant to evaluate the quality of the converted sentence.
    - Although the authors provided the instruction and actual evaluation process screenshot in the Appendix, they did not reveal the recruiting process of participants and their statistics, which makes the reliability of human evaluation results questionable due to potential selection bias [6].
  - The authors may refer to Section 2 of the GPQA paper [7], which elaborates on details about the human-in-the-loop dataset construction pipeline.

---
> Reference
1. Zheng, Lianmin, et al. "Judging llm-as-a-judge with mt-bench and chatbot arena." Advances in Neural Information Processing Systems 36 (2023): 46595-46623.
2. Dubois, Yann, et al. "Alpacafarm: A simulation framework for methods that learn from human feedback." Advances in Neural Information Processing Systems 36 (2023): 30039-30069.
3. Chiang, Wei-Lin, et al. "Chatbot arena: An open platform for evaluating llms by human preference." Forty-first International Conference on Machine Learning. 2024.
4. Jung, Haeji, et al. "Mitigating the Linguistic Gap with Phonemic Representations for Robust Cross-lingual Transfer." Proceedings of the Fourth Workshop on Multilingual Representation Learning (MRL 2024). 2024.
5. Mishra, Prabhaker; Singh, Uttam; Pandey, Chandra M; Mishra, Priyadarshni1; Pandey, Gaurav2. Application of Student's t-test, Analysis of Variance, and Covariance. Annals of Cardiac Anaesthesia 22(4):p 407-411, Oct–Dec 2019. | DOI: 10.4103/aca.ACA_94_19
6. Catania, Joseph A., et al. "Methodological problems in AIDS behavioral research: influences on measurement error and participation bias in studies of sexual behavior." Psychological bulletin 108.3 (1990): 339.
7. Rein, David, et al. "Gpqa: A graduate-level google-proof q&a benchmark." First Conference on Language Modeling. 2024.

**Strengths Contributions:**

* [`S1`] This work **focuses on a crucial yet (relatively) unexplored topic**, non-standard English performance analysis, in the era of LLMs.
  * The resulting datasets and the proposed framework, Trans-Env, can be flexibly leveraged to investigate the linguistic robustness of frontier-level language models and potentially contribute to making them safer.
* [`S2`] The authors proposed a solid framework that is **scalable** with LLM-based automatic transformations, while **supporting converted sentences' validity** via expert-driven transformation guidance rules and a semantic checker model.
  * This gentle incorporation of domain expertise and model-based automation results in a scalable and reliable pipeline to be adopted in real-world scenarios.

---

> ### Author Rebuttal · Authors · 2025-07-31
>
> **Experiments on Open-ended Tasks**
> We conducted experiments using three open-ended evaluation setups: IFEval, AlpacaFarm, and MT-Bench. We were unable to replicate all the experiments, selecting six English dialects (African American Vernacular English, Irish English, Australian English, Newfoundland English, Tristan da Cunha English, and Welsh English) and six ESL variants (Arabic, Italian, and Spanish with CEFR level A, B respectively). We evaluated three models: DeepSeek-R1-70B, LLaMA-3.3-70B, and Qwen-2.5-72B. We used an LLM-as-a-judge strategy, using GPT-4.1 as the evaluator.
>
> We translated the original instructions into the respective English variants and generated model outputs based on these transformed instructions. For IFEval, the metric is the accuracy of instruction adherence. For AlpacaFarm, we report the win rate when comparing model outputs against GPT-4. For MT-Bench, the evaluation LLM provides a score from 0 to 10 indicating how well the model's output aligns with the given instruction.
>
> |   Dataset  | Model           | Orig  | AAVE  | AuE   | TdCE  | IrE   | NFE   | WeE   |
> |:----------:|-----------------|-------|-------|-------|-------|-------|-------|-------|
> | IfEval     | Qwen-2.5-72B    | 0.824	 | 0.787 | 0.752 | 0.776	 | 0.695 | 0.704 | 0.721 |
> |            | DeepSeek-R1-70B | 0.765			 | 0.726	 | 0.710 | 0.723	 | 0.641 | 0.652	 | 0.665 |
> |            | LLaMA-3.3-70B   | 0.884		 | 0.837 | 0.824 | 0.845 | 0.758 | 0.758 | 0.776 |
> | AlpacaFARM | Qwen-2.5-72B    | 59.01 | 57.67 | 57.68 | 59.12	 | 57.36 | 59.56 | 58.56 |
> |            | DeepSeek-R1-70B | 46.78	 | 45.29 | 45.25 | 42.96	 | 45.51 | 44.56	 | 43.69 |
> |            | LLaMA-3.3-70B   | 39.77		 | 38.59 | 36.61 | 36.93 | 37.73	 | 36.87 | 36.57 |
> | MT-Bench   | Qwen-2.5-72B    |  9.04 |  8.89 |  9.10 |  8.91 |  8.60 |  8.52 |  8.93 |
> |            | DeepSeek-R1-70B |  8.63 |  8.66 |  8.93 |  8.46 |  8.30 |  8.22 |  8.76 |
> |            | LLaMA-3.3-70B   |  9.05 |  8.48 |  9.15 |  8.59 |  8.71 |  8.32 |  8.67 |
>
> ESL Result
> |   Dataset  | Model           | Orig  | A_ar  | A_it  | A_es  | B_ar  | B_it  | B_es  |
> |:----------:|-----------------|-------|-------|-------|-------|-------|-------|-------|
> | IfEval     | Qwen-2.5-72B            | 0.824 | 0.607 | 0.628 | 0.628 | 0.725 | 0.743 | 0.723 |
> |            | DeepSeek-R1-70B  | 0.765 | 0.597 | 0.574 | 0.592 | 0.656 | 0.664 | 0.641 |
> |            | LLaMA-3.3-70B       | 0.884 | 0.668 | 0.666 | 0.676 | 0.770 | 0.779 | 0.770 |
> | AlpacaFARM | Qwen-2.5-72B           | 59.01 | 53.91 | 54.49 | 55.33 | 54.34 | 56.82 | 58.03 |
> |            | DeepSeek-R1-70B  | 44.78 | 41.87 | 38.28 | 43.19 | 43.89 |  44.2 | 43.92 |
> |            | llama 70b       | 36.77 | 32.61 | 34.94 | 36.12 | 34.88 | 36.92 | 37.71 |
> | MT-Bench   | Qwen-2.5-72B            |  9.04 |  8.03 |  7.31 |  7.36 |  8.73 |  8.16 |  8.20 |
> |            | DeepSeek-R1-70B  |  8.63 |  7.94 |  7.22 |  7.11 |  8.20 |  7.81 |  7.26 |
> |            | LLaMA-3.3-70B       |  9.05 |  8.18 |  7.21 |  7.28 |  8.71 |  8.08 |  8.23 |
>
> Model performance declines across all three datasets when evaluated on the transformed instructions. Irish, Newfoundland, and Welsh exhibited lower performance compared to the others, mirroring trends observed in the closed-form evaluations. Similarly, ESL variants at CEFR level B outperformed those at level A. Additional experiments including evaluations with proprietary models and a broader range of English variants are ongoing. We will continue to update throughout the discussion period and include into the revised version of the paper.
>
> **Q2. Performance Gap Analysis with Sentence Level**
>
> Thank you for your insightful feedback. We have separated the six datasets based on word count and sentence count using four quantiles, and we report the average performance differences across models accordingly.
>
> | Dataset    | Word-Level  |       |       |       | Sentence-Level |       |       |       |
> |------------|-------------|-------|-------|-------|----------------|-------|-------|-------|
> |            | Q1          | Q2    | Q3    | Q4    | Q1             | Q2    | Q3    | Q4    |
> | MMLU       | 0.007       | 0.014 | 0.021 | 0.029 | 0.013          | 0.017 | 0.017 | 0.029 |
> | ARC        | 0.009       | 0.007 | 0.010 | 0.012 | 0.006          | 0.011 | 0.014 | 0.010 |
> | TruthfulQA | 0.014       | 0.017 | 0.013 | 0.020 | 0.010          | 0.011 | 0.013 | 0.054 |
> | GSM8k      | 0.026       | 0.028 | 0.018 | 0.014 | 0.027          | 0.018 | 0.027 | 0.012 |
> | hellswag   | 0.022       | 0.023 | 0.027 | 0.032 | 0.022          | 0.026 | 0.032 | 0.034 |
> | winogrande | 0.104       | 0.114 | 0.119 | 0.125 | 0.114          | 0.11  | 0.112 | 0.103 |
>
> All datasets, except for GSM8K, exhibit a linear trend in which performance degrades as sentence length increases, whether measured by word count or sentence count. We hypothesize that GSM8K does not follow this trend because its question lengths are relatively uniform. As a result, there are no substantial length differences across quantiles.
>
> **Q3. Figure 2 Correlation Value**
> Thank you for pointing out the missing details. We conducted three statistical tests, pearson correlation coefficient, spearman rank correlation, kendall’s tau, and the results are as below.
>
> | Dataset    | Pearson Correlation coefficient |          | Spearman Rank Correlation |          | Kendall’s Tau |          |
> |------------|---------------------------------|----------|---------------------------|----------|---------------|----------|
> |            | coefficient                     | p-value  | coefficient               | p-value  | coefficient   | p-value  |
> | MMLU       | 0.215                           | 0.025    | 0.193                     | 0.044    | 0.133         | 0.045    |
> | ARC        | 0.328                           | 0.0001   | 0.354                     | 4.67e-05 | 0.241         | 9.79e-05 |
> | TruthfulQA | 0.292                           | 0.0008   | 0.286                     | 0.001    | 0.198         | 0.001    |
> | GSM8k      | 0.321                           | 0.0002   | 0.226                     | 0.010    | 0.149         | 0.015    |
> | Hellaswag  | 0.118                           | 0.222    | 0.207                     | 0.030    | 0.141         | 0.033    |
> | WinoGrande | 0.429                           | 5.05e-07 | 0.438                     | 2.78e-07 | 0.271         | 1.04e-05 |
>
> The results show that all p-values, except for HellaSwag under the Pearson correlation coefficient, are below 0.05, indicating statistically significant linear correlations between performance gaps and linguistic distances.
>
> For ESL variants, we performed a one-way ANOVA to examine whether there are statistically significant differences among the four L1 groups. The results are presented below.
> | Dataset    | f_stat | p-value  |
> |------------|--------|----------|
> | MMLU       | 36.06  | 9.40e-10 |
> | ARC        | 92.43  | 1.23e-14 |
> | TruthfulQA | 25.91  | 3.12e-08 |
> | GSM8k      | 160.28 | 9.84e-18 |
> | Hellaswag  | 62.16  | 1.69e-12 |
> | WinoGrande | 43.67  | 1.08e-10 |
> Across all datasets, the p-values are extremely low, indicating clear performance differences among the four L1 groups.
>
> **Q. Usage of t-test in extracting L1 features**
>
> Thank you for the question. We chose the t-test to extract L1-specific linguistic features because our goal was to determine whether certain linguistic features appear significantly more frequently in one L1 corpus compared to others. Logistic regression and ANCOVA are indeed useful for assessing the importance of linguistic features. However, in logistic regression, the coefficient for each linguistic feature is estimated while other features are included in the model. As a result, it is difficult to interpret the independent effect of each feature. In particular, when there is a high correlation between features, the interpretation of the coefficients can become unstable. ANCOVA is appropriate when control variables are known a priori. In our case, the variables would be those linguistic features that appear in English learning regardless of the L1 background. Since we do not have prior knowledge of these variables, we did not apply ANCOVA in our analysis.
>
> **Q. Detailed explanation for explanation**
>
> In lines 120–121, We first described the full process of selecting the 18 English dialects, which was approved by the specialist. The specialist then compared the selected 18 dialects with the remaining non-selected dialects, evaluating their vocabulary, grammar, and other lexical characteristics. Based on this comparison, the specialist confirmed that the selected dialects were appropriate for the intended research purposes.
>
> In lines 156–157, We consulted the specialist to ensure the appropriateness of the experimental setup. After feature extraction, we presented the results to the specialist, who compared the extracted features against 13 well-established binary linguistic parameters known to characterize various languages. These parameters include Subject Position, Null Subject, Headedness in CP, and Preposition Stranding. We analyzed the correspondence between the extracted features and these parameters, and the specialist confirmed strong alignment between the linguistic parameters and our extracted features.
>
> **Q. Human Recruitment**
>
> We focused on recruiting participants with strong proficiency in English grammar, as the evaluation guidelines involved grammatical terminology and sentence structure analysis. We recruited graduate students who have had formal coursework in English linguistics or grammar. These candidates were asked to complete a grammar pre-test designed to assess their understanding of key concepts relevant to the evaluation task. Based on their performance, we selected three individuals who scored the highest. We will add these details in the revised paper.
>
> **Q. Captions**
>
> We will revise the caption and the figure color as you suggested.

---

> > ### Comment · Reviewer_E6DB · 2025-08-01
> >
> > I sincerely appreciate the authors' effort to provide additional results to make this benchmark more practical and persuasive.
> >
> > * My initial concern, "`W1`: Limited scope of validation" was addressed accordingly
> > * For the "`W3`: Lack of rigor in human study", the authors' clarification and plan for revision sound valid.
> >   * However, as the original manuscript did not have those details, I should not change any rating regarding this.
> > * For the "`W2`: Lack of rigor in statistical tests and their significance", the sub-item `W2-1` was addressed by the authors' additional experiments. Thanks for their hard work!!
> >   * However, it is still unclear to me **how they use t-test to extract L1-specific linguistic features**. Their current response does not contain the detailed procedure for the t-test for linguistic feature extraction. I would appreciate it if the author could explain this further.

---

> > > ### Author Response · Authors · 2025-08-01
> > > **Response to Reviewer E6DB**
> > >
> > > First, we sincerely thank the reviewer for their insightful comments.
> > >
> > >
> > > **[W3]** Thank you for the comments. We are glad that our comments clarify your questions regarding “W3: Lack of rigor in human study”. We acknowledge that these details should have been included in the original manuscript and it is our mistake that we did not include them in the original paper. We will make sure to include those details in the revised version.
> > >
> > >
> > > **[W2]** Thank you for your comment. Due to the response character limit, we were unable to present the full details of the t-test analysis. We employed the t-test to identify L1-specific linguistic features, as our objective was to determine whether particular features occur significantly more frequently in one L1 corpus compared to others. For example, to examine whether the “Omission of a Preposition” is characteristic of French speakers, we formulated the null hypothesis as: “The frequency of ‘Omission of a Preposition’ is the same in both the French and non-French corpora,” and the alternative hypothesis as: “The frequency of ‘Omission of a Preposition’ differs between the French and non-French corpora.”
> > >
> > >
> > > We would like to respectfully note that we have made every effort to address the concerns raised in your review within the limited time available. Specifically, we conducted three additional open-ended task experiments, analyzed the relationship between the performance gap and question length, and provided exact correlation values between the performance gap and linguistic feature distance. These were indeed valuable suggestions, and we sincerely acknowledge that the original manuscript lacked several important details. We will include all these analyses and clarifications in the revised version of the paper.
> > >
> > >
> > > We kindly ask that you take these points into consideration when finalizing your review, as some of the previously raised concerns, such as [W1], have been addressed. We hope our efforts are recognized in your overall evaluation. We believe that the purpose of the discussion phase is to clarify ambiguous aspects and allow for the additional information where appropriate, which may, in turn, justify adjustments to the review scores beyond what was initially presented in the original submission. If the information provided during the discussion period is not taken into account as it was not included in the original manuscript, our efforts in conducting additional experiments and analyses may unfortunately go unrecognized. We believe this would be contrary to the spirit and intended purpose of the NeurIPS review and discussion process. We kindly ask the reviewer to generously take the points raised in our rebuttal into consideration when determining the final review score.
> > >
> > >
> > > In line with the NeurIPS reviewer guidelines, which encourage reviewers to specify the conditions under which the evaluation score could be revised, we would greatly appreciate it if you could let us know what specific improvements might lead to a reconsideration of your evaluation. Additionally, if there are any other aspects that might influence your assessment, we would be grateful for your feedback.
> > >
> > >
> > > Thank you again for your time and thoughtful comments.

---

> > ### Comment · Reviewer_E6DB · 2025-08-01
> >
> > First of all, **I want to express my sincere APOLOGY** for making the authors feel anxious and stressed due to my ambiguous wording and bad communication. As an author of another submitted paper, I am also undergoing the same phase with the authors (doing tons of experiments with billion-scale LLM in a very limited time), so I deeply sympathize with the authors and how they feel pressured and stressed.
> > * **All of my concerns, except the t-test part, were addressed already through the initial response from the authors.**
> > * **But I just needed clarification on the details behind the t-test** (this was one of the most suspicious parts for me) before reconsidering my rating on this paper, because I have not seen any existing works that use the t-test for feature extraction/identification, if there are many papers on this approach, sorry for my lack of professionality.
> > * After reading the authors' faithful explanation on the t-test, my understanding of the procedure was improved, and I feel **the concern `W2-2` is also somewhat addressed**.
> > * Therefore, **I will raise my rating from 3 to 4** after finishing this writing!
> >
> > ---
> >
> > Meanwhile, allow me to clarify my statement about `W3`-- "_However, as the original manuscript did not have those details, I should not change any rating regarding this_".
> > * I did not mean that I will not change my rating on this paper on this sentence.
> > * I was just wanting to remind the fact that details lacking in the original draft, which are explained through this rebuttal thread, should not be assumed to have been originally included in the manuscript.
> > * I don't think the reviewer should solely judge the modification of rating based on the authors' promise to add details such as "[DESCRIPTION ON DETAILS]. We will add these details in the revised version".
> > * In contrast to the verbal commitment and promise on adding missing details, showing supporting evidence through additional empirical validation can change reviewers' opinion justifiably, because that evidence explicitly refutes reviewers' initial concern, even though it also did not appear in the original draft. The `W1` and `W2-1` are laid in this category.
> > * Moreover, for the `W2-2`, I needed clarification on how they use the t-test to precisely judge my rating, and the authors give sufficient clarification that fills my knowledge gap on that part, which can be a justification behind my reconsideration of the rating.
> >
> > ---
> >
> > In conclusion,
> > * I am deeply sorry about bordering the authors mentally (and also physically?). For the next review, I will be more professional in terms of communication by removing any aggressive sentences that would make the author nervous and being more specific. Thanks for giving me this lesson, and sorry again for your potential mental damage.
> > * Departing from it, I truly enjoyed reading and reviewing this paper, which I believe has huge potential to contribute robust and reliable LLM evaluation in a distinctive manner. So, good luck!
> >
> > ---
> > FYI, This year's NeurIPS system does not allow authors to access the updated rating and final justification upon the notification date, so maybe you still see the old rating (3), but I just changed it to 4 with a valid final justification.

---

> > > ### Author Response · Authors · 2025-08-02
> > >
> > > We sincerely thank you for your thoughtful reviews and the time you devoted to our discussions. It was a valuable opportunity for us to engage in such meaningful and high-quality exchanges. Your feedback has significantly contributed to improving the quality of our work. We are truly grateful that you recognized our efforts and recommended the paper for acceptance at NeurIPS. We feel exceptionally fortunate to have had you as our reviewer, as such in-depth, multi-turn discussions are rare and deeply appreciated in the review process.

---

> > > > ### Author Response · Authors · 2025-08-06
> > > >
> > > > Dear Reviewer E6DB,
> > > >
> > > > We are conducting an additional experiment regarding the Open-ended task (related to `w1-1`).
> > > > In this experiment, we include the o4-mini model, two dialects (Southeast England and North England), and two L1 backgrounds (French and Turkish).
> > > > The additional results are presented below.
> > > >
> > > > Dialect Result
> > > >
> > > > |   Dataset  |      Model      |  Orig |  AAVE |  AuE  |  TdCE |  IrE  |  NFE  |  WeE  | SE-Eng | NE-Eng |
> > > > |:----------:|:---------------:|:-----:|:-----:|:-----:|:-----:|:-----:|:-----:|:-----:|:------:|:------:|
> > > > |   IfEval   |   Qwen-2.5-72B  | 0.824 | 0.787 | 0.752 | 0.776 | 0.695 | 0.704 | 0.721 |  0.802 |  0.758 |
> > > > |            | DeepSeek-R1-70B | 0.765 | 0.726 | 0.710 | 0.723 | 0.641 | 0.652 | 0.665 |  0.713 |  0.699 |
> > > > |            |  LLaMA-3.3-70B  | 0.884 | 0.837 | 0.824 | 0.845 | 0.758 | 0.758 | 0.776 |  0.830 |  0.813 |
> > > > |            |     o4-mini     | 0.915 | 0.749 | 0.743 | 0.756 | 0.658 | 0.684 | 0.671 |  0.736 |  0.721 |
> > > > | AlpacaFARM |   Qwen-2.5-72B  | 59.01 | 57.67 | 57.68 | 59.12 | 57.36 | 59.56 | 58.56 |  57.86 |  54.70 |
> > > > |            | DeepSeek-R1-70B | 46.78 | 45.29 | 45.25 | 42.96 | 45.51 | 44.56 | 43.69 |  42.07 |  47.56 |
> > > > |            |  LLaMA-3.3-70B  | 39.77 | 38.59 | 36.61 | 36.93 | 37.73 | 36.87 | 36.57 |  38.19 |  36.63 |
> > > > |            |     o4-mini     | 84.00 | 40.86 | 39.79 | 41.15 | 40.82 | 41.07 | 40.77 |  40.20 |  41.25 |
> > > > |  MT-Bench  |   Qwen-2.5-72B  |  9.04 |  8.89 |  9.10 |  8.91 |  8.60 |  8.52 |  8.93 |  9.01  |  8.99  |
> > > > |            | DeepSeek-R1-70B |  8.63 |  8.66 |  8.93 |  8.46 |  8.30 |  8.22 |  8.76 |  8.58  |  8.56  |
> > > > |            |  LLaMA-3.3-70B  |  9.05 |  8.48 |  9.15 |  8.59 |  8.71 |  8.32 |  8.67 |  8.94  |  9.02  |
> > > > |            |     o4-mini     |  9.36 |  8.63 |  9.03 |  8.61 |  8.60 |  8.34 |  8.91 |  9.05  |  9.00  |
> > > >
> > > > ESL Result
> > > >
> > > > |   Dataset  |      Model      |  Orig |  A_ar |  A_it |  A_es |  A_fr |  A_tr |  B_ar |  B_it |  B_es |  B_fr |  B_tr |
> > > > |:----------:|:---------------:|:-----:|:-----:|:-----:|:-----:|:-----:|:-----:|:-----:|:-----:|:-----:|:-----:|:-----:|
> > > > |   IfEval   |   Qwen-2.5-72B  | 0.824 | 0.607 | 0.628 | 0.628 | 0.635 | 0.610 | 0.725 | 0.743 | 0.723 | 0.733 | 0.710 |
> > > > |            | DeepSeek-R1-70B | 0.765 | 0.597 | 0.574 | 0.592 | 0.559 | 0.564 | 0.656 | 0.664 | 0.641 | 0.624 | 0.642 |
> > > > |            |  LLaMA-3.3-70B  | 0.884 | 0.668 | 0.666 | 0.676 | 0.671 | 0.668 | 0.770 | 0.779 | 0.770 | 0.758 | 0.768 |
> > > > |            |     o4-mini     | 0.915 | 0.577 | 0.592 | 0.579 | 0.599 | 0.589 | 0.671 | 0.691 | 0.678 | 0.682 | 0.667 |
> > > > | AlpacaFARM |   Qwen-2.5-72B  | 59.01 | 53.91 | 54.49 | 55.33 | 57.11 | 56.95 | 54.34 | 56.82 | 58.03 | 56.77 | 58.52 |
> > > > |            | DeepSeek-R1-70B | 44.78 | 41.87 | 38.28 | 43.19 | 43.52 | 38.63 | 43.89 | 44.20 | 43.92 | 41.99 | 40.31 |
> > > > |            |  LLaMA-3.3-70B  | 36.77 | 32.61 | 34.94 | 36.12 | 36.00 | 36.02 | 34.88 | 36.92 | 37.71 | 37.19 | 36.71 |
> > > > |            |     o4-mini     | 84.00 | 76.74 | 77.57 | 78.76 | 81.30 | 81.07 | 77.35 | 80.88 | 82.60 | 80.81 | 83.30 |
> > > > |  MT-Bench  |   Qwen-2.5-72B  |  9.04 |  8.03 |  7.31 |  7.36 |  7.36 |  7.51 |  8.73 |  8.16 |  8.20 |  7.99 |  8.29 |
> > > > |            | DeepSeek-R1-70B |  8.63 |  7.94 |  7.22 |  7.11 |  7.32 |  7.08 |  8.20 |  7.81 |  7.26 |  7.60 |  7.44 |
> > > > |            |  LLaMA-3.3-70B  |  9.05 |  8.18 |  7.21 |  7.28 |  7.31 |  7.20 |  8.71 |  8.08 |  8.23 |  8.36 |  8.28 |
> > > > |            |     o4-mini     |  9.36 |  8.13 |  7.48 |  7.19 |  7.43 |  7.47 |  8.45 |  8.36 |  8.43 |  8.16 |  8.21 |
> > > >
> > > > Consistent with the results observed for the existing English variants and models, the additional experiments also demonstrate degraded performance compared to the SAE instruction.
> > > >
> > > > We thank the reviewer for suggesting additional experiments on the open-ended tasks, and we will incorporate these results into the final revised paper.

---

> > > > > ### Comment · Reviewer_E6DB · 2025-08-09
> > > > >
> > > > > Dear Authors,
> > > > >
> > > > > Thank you so much for providing those additional validation results.
> > > > >
> > > > > As my concern `W1` was already well-addressed by your initial rebuttal, and I have reflected that fact on my rating update a few days ago, there will be no further explicit change. But I feel the comprehensiveness of this benchmark is further increased by the additional results on o4-mini and dialects/L1. I will reflect this improved positive impression in this paper during the AC-reviewer discussion phase, if needed. Thanks.
> > > > >
> > > > > However, departing from the rating or the acceptance of this work, the authors' attitude toward this submission is very impressive (thoughtful, active, and professional). It was my pleasure to review this paper.
> > > > >
> > > > > Best,
> > > > >
> > > > > Reviewer E6DB

---

### Official Review · Reviewer_5sPe · 2025-07-01

**Rating:** 5
**Confidence:** 4

**Summary:**

This paper introduces Trans-EnV, a novel and scalable framework for evaluating the linguistic robustness of Large Language Models (LLMs) against a wide array of English varieties. The authors identify a critical gap in current LLM evaluation, which is predominantly focused on Standard American English (SAE), leading to fairness and robustness concerns.

**Dataset Code Accessibility:**

Yes

**Dataset Code Comments:**

1. For the ESL English generation, a 15% threshold for higher-level vocabulary was used to simulate realistic proficiency. This is well-justified by analyzing a CEFR-labeled corpus. Did you perform any sensitivity analysis on this hyperparameter? How would a stricter (e.g., 5%) or more lenient (e.g., 25%) threshold affect model performance?

2. The choice of LLaMA-3.3-70B as the semantic checker is interesting. How much did this checker contribute to the final quality?

**Ethical Considerations:**

No, there are no or only very minor ethics concerns

**Limitations Weaknesses:**

The quality of the generated datasets is inherently tied to the capabilities of the chosen feature transformer (Gemma-2-27B) and semantic checker (LLaMA-3.3-70B). A brief discussion in the limitations section about the framework's sensitivity to this choice and how the quality might degrade with less capable models would be beneficial.

**Strengths Contributions:**

1. The paper tackles a crucial and increasingly relevant problem of fairness and robustness in LLMs as they become globally deployed. Creating a systematic and scalable benchmark for English varieties is a high-impact contribution for the NLP and ML communities.

2. The core strength of Trans-EnV lies in its rigorous methodology. Instead of relying on naive prompting for style transfer, it grounds the transformation process in expert-curated linguistic databases. This "expert-in-the-loop" approach, where linguistic knowledge guides LLM-driven execution, ensures the generated data is linguistically plausible and valid, a feature often missing in prior work.

3. The experimental setup is extensive, covering 38 varieties, 6 diverse tasks (knowledge-based and reasoning), and 7 modern LLMs.

---

> ### Author Rebuttal · Authors · 2025-07-31
>
> **Q. Vocabulary Threshold**
> Thank you for your suggestion. We conducted additional experiments using TruthfulQA with LLaMA-3.3-70B-Instruct. Due to time constraints, we were unable to evaluate all ESL variants. The results are presented below:
>
> |     | A_arabic | A_italian | B_arabic | B_italian |
> |-----|----------|-----------|----------|-----------|
> | 15% (Reported in the paper) |     63.2 |      65.3 |       63 |      66.8 |
> |  25% |     63.8 |      66.8 |     65.3 |      67.4 |
> | 5% |     60.4 |      63.2 |     62.1 |      64.2 |
>
> The 15% threshold was used in the main paper. As shown in the table, a stricter threshold (5%) leads to a decline in performance, whereas a more lenient threshold (25%) results in improved performance across all variants. We hypothesize that a stricter threshold compels the model to replace a greater number of vocabulary items with simpler alternatives, thereby increasing the degree of transformation from the original sentence and potentially compromising meaning or coherence.
>
> We thank you for your insightful feedback and will include this analysis in the final version of the paper.
>
>
> **Q. choice of LLaMA-3.3-70B as semantic checker**
> When using Gemma-2-27B-Instruct as the feature transformation model, we observed instances where portions of the original questions were omitted in the transformed outputs. For example, given the original question “A man stands in an elevator as it begins to ascend. Does the normal force from the floor do work on the man?”, the initial transformed version was “The, like, normal force from the floor do work on the man?”, which omits key contextual information. In such cases, LLaMA-3.3-70B, serving as the semantic consistency checker, successfully identified the semantic misalignment.
>
> Upon regenerating these cases, approximately 78% of the outputs retained the original meaning more accurately. For instance, the revised transformation “A man stands in an elevator as it begins to ascend. Does, like, the normal force from the floor do work on the man?” preserved the intended semantics while incorporating the target linguistic feature.

---

> ### Comment · Area_Chair_LHBV · 2025-08-06
>
> Dear Reviewer 5sPe,
>
> Would you check the author's reply and start the discussion based on it? I want to gently remind you that, according to the NeurIPS reviewing policy, Reviewers must participate in discussions with authors before submitting the “Mandatory Acknowledgement”.
>
> Thank you for your great support!
>
> Best wishes,
>
> AC

---

### Official Review · Reviewer_e6nB · 2025-07-03

**Rating:** 5
**Confidence:** 3

**Summary:**

This paper introduces Trans-EnV, an automated framework that transforms Standard American English (SAE) datasets into various English varieties to evaluate the linguistic robustness of LLMs. The framework combines linguistic expert knowledge from curated resources (eWAVE database, English Grammar Profile, and ESL learner corpora) with LLM-based transformations to create evaluation datasets across 38 English varieties (18 dialects and 20 ESL English varieties). The authors transform six benchmark QA datasets and evaluate seven state-of-the-art LLMs, revealing performance drops of up to 12.5% for dialects and 46.3% for ESL English varieties compared to SAE.

**Additional Feedback:**

It would be helpful if the paper could mention the computational requirements for transformation, especially when scaling to larger datasets or stronger models.

**Dataset Code Accessibility:**

Yes

**Ethical Considerations:**

No, there are no or only very minor ethics concerns

**Final Justification:**

The authors provide detailed response and additional results, which resolved most of my concerns. Hence, I will increase my score.

**Limitations Weaknesses:**

1. The evaluation is restricted to QA tasks only. The generalizability to other important NLP tasks like generation, summarization, or dialogue remains unexplored, limiting the scope of conclusions about overall LLM robustness.
2. While the framework was validated by a second language acquisition researcher, there's limited validation from native speakers of the target varieties (only Spanish and French were mentioned). This could affect the authenticity of transformations.

**Strengths Contributions:**

1. The paper tackles a important fairness issue in NLP - the performance disparity of LLMs on non-standard English varieties, which affects global users who do not speak SAE.
2. Trans-EnV is built on solid linguistic foundations, utilizing expert-curated resources like eWAVE (compiled by 84 professional linguists) and the English Grammar Profile. The framework was developed in consultation with a researcher specializing in second language acquisition, ensuring linguistic validity.
3. The framework combines the scalability of LLMs with the accuracy of expert-curated linguistic features, addressing limitations of purely manual or purely LLM-based approaches.
4. The paper presents thorough experiments across 38 varieties, 6 benchmark datasets, and 7 sota models, providing compelling evidence of the robustness issues. The human evaluation (88% validity rate) and semantic checker validation demonstrate the quality of transformations.

---

> ### Author Rebuttal · Authors · 2025-07-31
>
> **Q. Additional Experiments on Open-ended Tasks**
>
> Thank you for your valuable suggestion. We conducted additional experiments using three open-ended evaluation setups: IFEval, AlpacaFarm, and MT-Bench. Due to time constraints, we were unable to replicate all the experiments reported in the original paper. Instead, we selected six English dialects (African American Vernacular English, Irish English, Australian English, Newfoundland English, Tristan da Cunha English, and Welsh English) and six ESL variants (Arabic, Italian, and Spanish with CEFR level A, B respectively). These were chosen to include both relatively strong and weak performance categories.
>
> 	We evaluated three models: DeepSeek-R1-70B, LLaMA-3.3-70B, and Qwen-2.5-72B. While an ideal evaluation would include human judgments, practical limitations in time and resources led us to adopt an LLM-as-a-judge strategy, using GPT-4.1 as the evaluation model in line with prior work.
>
> 	For each dataset, we translated the original instructions into the respective English variants and generated model outputs based on these transformed instructions. The results are summarized in the table below. For IFEval, the metric is the accuracy of instruction adherence. For AlpacaFarm, we report the win rate when comparing model outputs against GPT-4. For MT-Bench, the evaluation LLM provides a score from 0 to 10 indicating how well the model's output aligns with the given instruction.
>
> Dialect Result
> |   Dataset  | Model           | Orig  | AAVE  | AuE   | TdCE  | IrE   | NFE   | WeE   |
> |:----------:|-----------------|-------|-------|-------|-------|-------|-------|-------|
> | IfEval     | Qwen-2.5-72B    | 0.824	 | 0.787 | 0.752 | 0.776	 | 0.695 | 0.704 | 0.721 |
> |            | DeepSeek-R1-70B | 0.765			 | 0.726	 | 0.710 | 0.723	 | 0.641 | 0.652	 | 0.665 |
> |            | LLaMA-3.3-70B   | 0.884		 | 0.837 | 0.824 | 0.845 | 0.758 | 0.758 | 0.776 |
> | AlpacaFARM | Qwen-2.5-72B    | 59.01 | 57.67 | 57.68 | 59.12	 | 57.36 | 59.56 | 58.56 |
> |            | DeepSeek-R1-70B | 46.78	 | 45.29 | 45.25 | 42.96	 | 45.51 | 44.56	 | 43.69 |
> |            | LLaMA-3.3-70B   | 39.77		 | 38.59 | 36.61 | 36.93 | 37.73	 | 36.87 | 36.57 |
> | MT-Bench   | Qwen-2.5-72B    |  9.04 |  8.89 |  9.10 |  8.91 |  8.60 |  8.52 |  8.93 |
> |            | DeepSeek-R1-70B |  8.63 |  8.66 |  8.93 |  8.46 |  8.30 |  8.22 |  8.76 |
> |            | LLaMA-3.3-70B   |  9.05 |  8.48 |  9.15 |  8.59 |  8.71 |  8.32 |  8.67 |
>
> ESL Result
> |   Dataset  | Model           | Orig  | A_ar  | A_it  | A_es  | B_ar  | B_it  | B_es  |
> |:----------:|-----------------|-------|-------|-------|-------|-------|-------|-------|
> | IfEval     | Qwen-2.5-72B            | 0.824 | 0.607 | 0.628 | 0.628 | 0.725 | 0.743 | 0.723 |
> |            | DeepSeek-R1-70B  | 0.765 | 0.597 | 0.574 | 0.592 | 0.656 | 0.664 | 0.641 |
> |            | LLaMA-3.3-70B       | 0.884 | 0.668 | 0.666 | 0.676 | 0.770 | 0.779 | 0.770 |
> | AlpacaFARM | Qwen-2.5-72B           | 59.01 | 53.91 | 54.49 | 55.33 | 54.34 | 56.82 | 58.03 |
> |            | DeepSeek-R1-70B  | 44.78 | 41.87 | 38.28 | 43.19 | 43.89 |  44.2 | 43.92 |
> |            | llama 70b       | 36.77 | 32.61 | 34.94 | 36.12 | 34.88 | 36.92 | 37.71 |
> | MT-Bench   | Qwen-2.5-72B            |  9.04 |  8.03 |  7.31 |  7.36 |  8.73 |  8.16 |  8.20 |
> |            | DeepSeek-R1-70B  |  8.63 |  7.94 |  7.22 |  7.11 |  8.20 |  7.81 |  7.26 |
> |            | LLaMA-3.3-70B       |  9.05 |  8.18 |  7.21 |  7.28 |  8.71 |  8.08 |  8.23 |
>
> Consistent with the results observed in closed-form tasks, model performance declines across all three datasets when evaluated on the transformed instructions. Notably, Irish English, Newfoundland English, and Welsh English exhibited lower performance compared to African American Vernacular English, Australian English, and Tristan da Cunha English, mirroring trends observed in the closed-form evaluations. Similarly, ESL variants at CEFR level B outperformed those at level A, reflecting a pattern consistent with earlier findings.
>
> Although we were only able to present partial experimental results due to time constraints, additional experiments are ongoing. These include evaluations with proprietary models such as o4-mini and the inclusion of a broader range of English variants. We will continue to provide updates throughout the discussion period and incorporate the extended results into the revised version of the paper.
>
> **Q. Native Speaker Validation**
>
> Thank you for pointing this out. We fully agree that validation by native speakers of a broader range of target varieties would enhance the authenticity of the transformations. However, due to logistical constraints, we were unable to extend the validation process to additional languages. Specifically, we needed to ensure that participants were both highly proficient in English and native speakers of the target languages, and we conducted in-depth interviews with them in person.
>
> In addition to the validation presented in the main paper, we also interviewed native speakers of Chinese and Italian. Following a similar procedure, we presented participants with the extracted features and asked them to determine whether these reflected common English errors made by native speakers of their language. Participants were also asked to explain the connection between each feature and characteristics of their native language. Both the Chinese and Italian participants confirmed that the majority of the extracted features aligned with common English errors among speakers of their respective languages.
>
> We acknowledge that involving a wider range of native speakers would have provided a more comprehensive validation. We consider broader native speaker engagement an important avenue for future research.
>
> **Q. Computation Requirements**
> Thank you for pointing out the computation requirements.
> For our case, we used 2 NVIDIA RTX A6000 GPU to transform each dataset into targeted variants using Gemma-2-27B-Instruct.
> Below are the time required for transformation for each dataset.
>
> - MMLU: 11 hours
> - ARC: 2h 50 min
> - TruthfulQA: 1h 50min
> - GSM8k: 3h
> - Hellaswag: 8h
> - Winogrande: 2h 40min

---

> > ### Comment · Reviewer_e6nB · 2025-08-04
> >
> > Thanks for the additional results. They have resolved most of my concerns, and I will increase the score.

---

> > > ### Author Response · Authors · 2025-08-05
> > >
> > > We sincerely are grateful that the reviewer recognized the novelty and contribution of our paper. Your feedback indeed helped improve our work, and it was a pleasure to engage in discussion about our paper.
> > >
> > > Thank you again for your time and valuable comments. We are happy to have had you as a reviewer.

---

> > > > ### Author Response · Authors · 2025-08-06
> > > >
> > > > Dear Reviewer e6nB,
> > > >
> > > > We are conducting an additional experiment regarding the Open-ended task.
> > > > In this experiment, we include the o4-mini model, two dialects (Southeast England and North England), and two L1 backgrounds (French and Turkish).
> > > > The additional results are presented below.
> > > >
> > > > Dialect Result
> > > > |   Dataset  |      Model      |  Orig |  AAVE |  AuE  |  TdCE |  IrE  |  NFE  |  WeE  | SE-Eng | NE-Eng |
> > > > |:----------:|:---------------:|:-----:|:-----:|:-----:|:-----:|:-----:|:-----:|:-----:|:------:|:------:|
> > > > |   IfEval   |   Qwen-2.5-72B  | 0.824 | 0.787 | 0.752 | 0.776 | 0.695 | 0.704 | 0.721 |  0.802 |  0.758 |
> > > > |            | DeepSeek-R1-70B | 0.765 | 0.726 | 0.710 | 0.723 | 0.641 | 0.652 | 0.665 |  0.713 |  0.699 |
> > > > |            |  LLaMA-3.3-70B  | 0.884 | 0.837 | 0.824 | 0.845 | 0.758 | 0.758 | 0.776 |  0.830 |  0.813 |
> > > > |            |     o4-mini     | 0.915 | 0.749 | 0.743 | 0.756 | 0.658 | 0.684 | 0.671 |  0.736 |  0.721 |
> > > > | AlpacaFARM |   Qwen-2.5-72B  | 59.01 | 57.67 | 57.68 | 59.12 | 57.36 | 59.56 | 58.56 |  57.86 |  54.70 |
> > > > |            | DeepSeek-R1-70B | 46.78 | 45.29 | 45.25 | 42.96 | 45.51 | 44.56 | 43.69 |  42.07 |  47.56 |
> > > > |            |  LLaMA-3.3-70B  | 39.77 | 38.59 | 36.61 | 36.93 | 37.73 | 36.87 | 36.57 |  38.19 |  36.63 |
> > > > |            |     o4-mini     | 84.00 | 40.86 | 39.79 | 41.15 | 40.82 | 41.07 | 40.77 |  40.20 |  41.25 |
> > > > |  MT-Bench  |   Qwen-2.5-72B  |  9.04 |  8.89 |  9.10 |  8.91 |  8.60 |  8.52 |  8.93 |  9.01  |  8.99  |
> > > > |            | DeepSeek-R1-70B |  8.63 |  8.66 |  8.93 |  8.46 |  8.30 |  8.22 |  8.76 |  8.58  |  8.56  |
> > > > |            |  LLaMA-3.3-70B  |  9.05 |  8.48 |  9.15 |  8.59 |  8.71 |  8.32 |  8.67 |  8.94  |  9.02  |
> > > > |            |     o4-mini     |  9.36 |  8.63 |  9.03 |  8.61 |  8.60 |  8.34 |  8.91 |  9.05  |  9.00  |
> > > >
> > > > ESL Result
> > > > |   Dataset  |      Model      |  Orig |  A_ar |  A_it |  A_es |  A_fr |  A_tr |  B_ar |  B_it |  B_es |  B_fr |  B_tr |
> > > > |:----------:|:---------------:|:-----:|:-----:|:-----:|:-----:|:-----:|:-----:|:-----:|:-----:|:-----:|:-----:|:-----:|
> > > > |   IfEval   |   Qwen-2.5-72B  | 0.824 | 0.607 | 0.628 | 0.628 | 0.635 | 0.610 | 0.725 | 0.743 | 0.723 | 0.733 | 0.710 |
> > > > |            | DeepSeek-R1-70B | 0.765 | 0.597 | 0.574 | 0.592 | 0.559 | 0.564 | 0.656 | 0.664 | 0.641 | 0.624 | 0.642 |
> > > > |            |  LLaMA-3.3-70B  | 0.884 | 0.668 | 0.666 | 0.676 | 0.671 | 0.668 | 0.770 | 0.779 | 0.770 | 0.758 | 0.768 |
> > > > |            |     o4-mini     | 0.915 | 0.577 | 0.592 | 0.579 | 0.599 | 0.589 | 0.671 | 0.691 | 0.678 | 0.682 | 0.667 |
> > > > | AlpacaFARM |   Qwen-2.5-72B  | 59.01 | 53.91 | 54.49 | 55.33 | 57.11 | 56.95 | 54.34 | 56.82 | 58.03 | 56.77 | 58.52 |
> > > > |            | DeepSeek-R1-70B | 44.78 | 41.87 | 38.28 | 43.19 | 43.52 | 38.63 | 43.89 | 44.20 | 43.92 | 41.99 | 40.31 |
> > > > |            |  LLaMA-3.3-70B  | 36.77 | 32.61 | 34.94 | 36.12 | 36.00 | 36.02 | 34.88 | 36.92 | 37.71 | 37.19 | 36.71 |
> > > > |            |     o4-mini     | 84.00 | 76.74 | 77.57 | 78.76 | 81.30 | 81.07 | 77.35 | 80.88 | 82.60 | 80.81 | 83.30 |
> > > > |  MT-Bench  |   Qwen-2.5-72B  |  9.04 |  8.03 |  7.31 |  7.36 |  7.36 |  7.51 |  8.73 |  8.16 |  8.20 |  7.99 |  8.29 |
> > > > |            | DeepSeek-R1-70B |  8.63 |  7.94 |  7.22 |  7.11 |  7.32 |  7.08 |  8.20 |  7.81 |  7.26 |  7.60 |  7.44 |
> > > > |            |  LLaMA-3.3-70B  |  9.05 |  8.18 |  7.21 |  7.28 |  7.31 |  7.20 |  8.71 |  8.08 |  8.23 |  8.36 |  8.28 |
> > > > |            |     o4-mini     |  9.36 |  8.13 |  7.48 |  7.19 |  7.43 |  7.47 |  8.45 |  8.36 |  8.43 |  8.16 |  8.21 |
> > > >
> > > > In line with the trends observed in the previously tested English variants and models, the supplementary experiments likewise indicate a decline in performance relative to the SAE instruction.
> > > >
> > > > We thank the reviewer’s recommendation to conduct additional evaluations on the open-ended tasks, and these new findings will be incorporated into the final revised manuscript.

---

### Note · Authors · 2025-08-12

We sincerely thank the reviewers, ACs, and SACs for the time and effort they dedicated to our work. This process has been a valuable opportunity for us to engage in meaningful and constructive discussions. To aid in the final decision, we briefly summarize our contributions and the outcomes of the discussion.

Our work introduces **Trans-Env**, a framework that automatically transforms Standard American English (SAE) datasets into multiple English varieties to evaluate the robustness of large language models (LLMs) to linguistic variation. The framework is grounded in expert-curated linguistic resources, and its construction was validated through rigorous statistical testing and consultation with a linguist, ensuring linguistic validity. Through extensive experiments with seven LLMs across six benchmarks, we demonstrate that LLM performance is impacted by linguistic changes.

The main initial concern in the reviews was that our experiments were limited to closed-ended tasks. To address this, we conducted additional experiments on open-ended tasks across three datasets (IfEval, AlpacaFARM, and MT-Bench) with four models on eight dialects and ten ESL variants. These results confirm that LLMs still exhibit degraded performance compared to SAE, underscoring the necessity of our transformation framework. Furthermore, during the rebuttal period, we made our best effort to address all concerns by conducting additional validations with two native speakers and providing further clarifications and analyses.

We are pleased that most concerns raised by the reviewers have been addressed, as reflected in their comments: *“They have resolved most of my concerns” (Reviewer e6nB)*, *“My initial concern was addressed accordingly, the authors’ clarification and plan for revision sound valid,” and “W2-1 was addressed by the authors’ additional experiments” (Reviewer E6DB)*. We are also grateful for the recognition of our efforts during the discussion period, with remarks noting that *“the authors’ attitude toward this submission is very impressive (thoughtful, active, and professional).” (Reviewer E6DB)*

We deeply appreciate the opportunity to present and discuss our work, and we thank the reviewers and chairs once again for their thoughtful engagement and valuable feedback.

---

### Decision · Program_Chairs · 2025-09-18

**Decision:**

Accept (poster)

**Comment:**

The paper introduces Trans-EnV, a framework for evaluating the linguistic robustness of large language models (LLMs) against English varieties. The framework automatically transforms Standard American English datasets into 38 English varieties (18 dialects and 20 ESL variants) by combining expert-curated linguistic rules with LLM-based transformations. Evaluations across six benchmark datasets and seven state-of-the-art LLMs show substantial performance disparities.

Comprehensively, this paper makes a timely, high-impact contribution by introducing the first scalable framework to systematically evaluate LLMs on diverse English varieties. Despite some initial concerns about task scope and validation rigor, the authors addressed them thoroughly during rebuttal with additional experiments and clarifications. The combination of expert-driven methodology, extensive empirical evidence, and community resources ensures strong scientific merit and long-term value. We believe this work directly advances fairness and robustness in LLM evaluation, making it suitable for acceptance at NeurIPS 2025.